# Skeletal muscle transcriptome in healthy aging

Robert A. Tumasian III [1], Abhinav Harish[1], Gautam Kundu[1], Jen-Hao Yang[1], Ceereena Ubaida-Mohien[1], Marta Gonzalez-Freire[1], Mary Kaileh[1], Linda M. Zukley[1], Chee W. Chia[1], Alexey Lyashkov[1], William H. Wood III[1], Yulan Piao[1], Christopher Coletta[1], Jun Ding[1], Myriam Gorospe[1], Ranjan Sen[1], Supriyo De[1] & Luigi Ferrucci [1]✉

Age-associated changes in gene expression in skeletal muscle of healthy individuals reflect accumulation of damage and compensatory adaptations to preserve tissue integrity. To characterize these changes, RNA was extracted and sequenced from muscle biopsies collected from 53 healthy individuals (22–83 years old) of the GESTALT study of the National Institute on Aging–NIH. Expression levels of 57,205 protein-coding and non-coding RNAs were studied as a function of aging by linear and negative binomial regression models. From both models, 1134 RNAs changed significantly with age. The most differentially abundant mRNAs encoded proteins implicated in several age-related processes, including cellular senescence, insulin signaling, and myogenesis. Specific mRNA isoforms that changed significantly with age in skeletal muscle were enriched for proteins involved in oxidative phosphorylation and adipogenesis. Our study establishes a detailed framework of the global transcriptome and mRNA isoforms that govern muscle damage and homeostasis with age.

[1] National Institute on Aging–Intramural Research Program, National Institutes of Health, Baltimore, MD, USA. ✉email: FerrucciLu@grc.nia.nih.gov

As older populations continue to expand worldwide, there is growing urgency to discern the underlying biological mechanisms that drive the deleterious manifestations of aging, including disease susceptibility and functional impairments. The study of biomarkers associated with healthy aging in the absence of overt disease may help uncover mechanisms of biological aging, including the accumulation of damage and intervening homeostatic mechanisms. A first step in this process is to produce detailed catalogs of molecular parameters that change systematically over the lifespan in a population of very healthy individuals.

The Genetic and Epigenetic Signatures of Translational Aging Laboratory Testing (GESTALT) study of the National Institute on Aging–National Institutes of Health (NIH) was designed to collect and analyze epigenetic, transcriptomic, and proteomic biomarkers from multiple tissues (including the muscle) from healthy individuals dispersed over a wide age range[1]. The specific goal of the study reported here is to elucidate differences in the transcriptomic network of the skeletal muscle as a function of age. Earlier large-scale studies identified changes in gene expression associated with skeletal muscle aging and acute exercise in well-defined populations using microarrays[2,3]. Other studies have also identified genomic and proteomic signatures associated with skeletal muscle aging[4–7]. However, no previous study has collected this information from very healthy individuals over a wide age range enrolled according to strict clinical and functional criteria.

To overcome some of these limitations, and to replicate and complement information collected in previous studies, we performed an RNA-sequencing (RNA-seq) analysis from skeletal muscle biopsies gathered from 53 healthy individuals of ages ranging from 22 to 83 years old. We used the data to comprehensively analyze changes in gene expression and isoforms that occur with aging. Some of our results identify basic biological mechanisms of aging, such as cellular senescence and insulin signaling. Other results confirm previous findings in the literature, in some cases, for genes whose function in the muscle and in aging is not well understood. Elucidation of the mechanisms that drive changes in the expression of these genes with aging and a stronger knowledge of their physiological functions are critical for understanding both damage accumulation as well as strategies of biological resilience with aging, which could serve as targets for interventions. We postulate that imbalances in these same mechanisms and compensatory strategies are likely to cause debilitating pathologies of aging muscle, including frailty and sarcopenia.

## Results

**Participant characteristics.** Skeletal muscle biopsies were obtained from 53 very healthy GESTALT participants (22–83 years, median = 52 years), who were defined as very healthy based on strict inclusion criteria developed by the Clinical Research Unit of the National Institute on Aging[8] (see "Methods"). The majority of our cohort was Caucasian ($n = 38$) and male ($n = 33$), and completed at least a high school education ($n = 30$) (Supplementary Fig. 1a). In spite of the strict inclusion criteria, there were significant ($p < 0.05$) differences in waist circumference (cm), fitness (VO₂ max, ml/kg/min), fasting glucose (mg/dL), 400 m walking time (s), and knee strength (Nm) between younger (≤52 years) and older (>52 years) participants (Supplementary Fig. 1a).

**RNA (human Ensembl genes (ENSG)) and isoform (human Ensembl transcripts (ENST)) detection.** Prior to studying age-related changes in the human skeletal muscle transcriptome, we determined the average number of protein-coding and noncoding RNAs (ENSGs) that were reliably detected using the Ensembl hg19 v82 (September 2015) database. On average, 24,453 (ranging from 16,203 to 36,119) RNAs were detected per sample in our study population, among which 15,291 (ranging from 11,517 to 18,966) were protein-coding RNAs and 9162 (ranging from 4686 to 17,339) were noncoding and other types of RNAs (Supplementary Fig. 1b). By "other types," we mean the biotypes that are not specifically mentioned as bona fide "protein-coding," and which cannot be categorized well. Similarly, on average, 66,633 (ranging from 42,928 to 110,124) mRNA isoforms were detected per sample in our study population, among which 27,908 (ranging from 20,954 to 40,371) were protein-coding isoforms and 38,075 (ranging from 21,768 to 69,753) were noncoding and other types of isoforms (Supplementary Fig. 1c). Overall, roughly 85% of all detected RNAs and 84% of all detected isoforms were identified in the Ensembl hg19 v82 (September 2015) database, underscoring the reliability of our data (Supplementary Fig. 1b, c). In this study, we define ENSGs as genes/RNAs and ENSTs as transcripts/isoforms.

**Analysis of RNA (ENSG) expression profiles in GESTALT participants.** An overview of the sample preparation and analytical approaches used in this study is provided (Fig. 1a). Linear regression analysis identified 664 RNAs that were overrepresented ($p < 0.01$, positive $\beta$-values for age) and 57 RNAs that were underrepresented ($p < 0.01$, negative $\beta$-values for age) in older compared to younger individuals (Fig. 1b). Together, the standard and zero-adjusted negative binomial models (see "Methods") identified 854 RNAs that were overrepresented and 65 RNAs that were underrepresented with older age (Fig. 1b). Of note, the directionality of RNA age associations was highly consistent between the two models. Overall, 506 RNAs were significantly ($p < 0.01$) associated with age from both the linear and negative binomial models; of these, 483 RNAs had higher expression with older age and 23 RNAs had lower expression with older age (Fig. 1c). Moreover, eight of the top ten RNAs with the lowest $p$-values overlapped between the two models, underscoring the robustness of these findings. A heat map of the expression levels of these 506 shared RNAs with age is shown (Fig. 1d), with the 23 underrepresented RNAs clustered at the bottom (blue to yellow rows with increasing age). Only a 37-year-old donor and an 80-year-old donor did not cluster with the study population. As the medical records of these participants did not suggest any possible cause for exclusion and the overall trends of expression level with age were conserved, all data for these participants were maintained in our analysis.

**Top RNAs (ENSGs) displaying differential gene expression in the skeletal muscle with age.** For in-depth analysis and validation, we focused on the top 20 RNAs most related to aging, ranked by the lowest $p$-value, which included a few noncoding RNAs, including pseudogene-encoded RNAs and antisense RNAs. The names, biotypes (protein-coding or noncoding), associated functions, and $\beta$-values for age identified by the linear model are listed (Figs. 2a and 3a). Linear regression plots for the top ten RNAs within these groups with positive and negative $\beta$-values for age are shown in Figs. 2b and 3b, respectively.

Our analysis found age-associated transcripts that have never been reported in the literature. Specifically, cyclin-dependent kinase inhibitor 2B (*CDKN2B*) mRNA (cyclin-dependent kinase inhibitor p15INK4b, which arrests the cell cycle by inhibiting cyclin-dependent kinase (CDK) 4 activity and is implicated in cellular senescence[9]) was the top age-related transcript, followed by *IRS2* mRNA (insulin receptor substrate 2, involved in insulin signaling

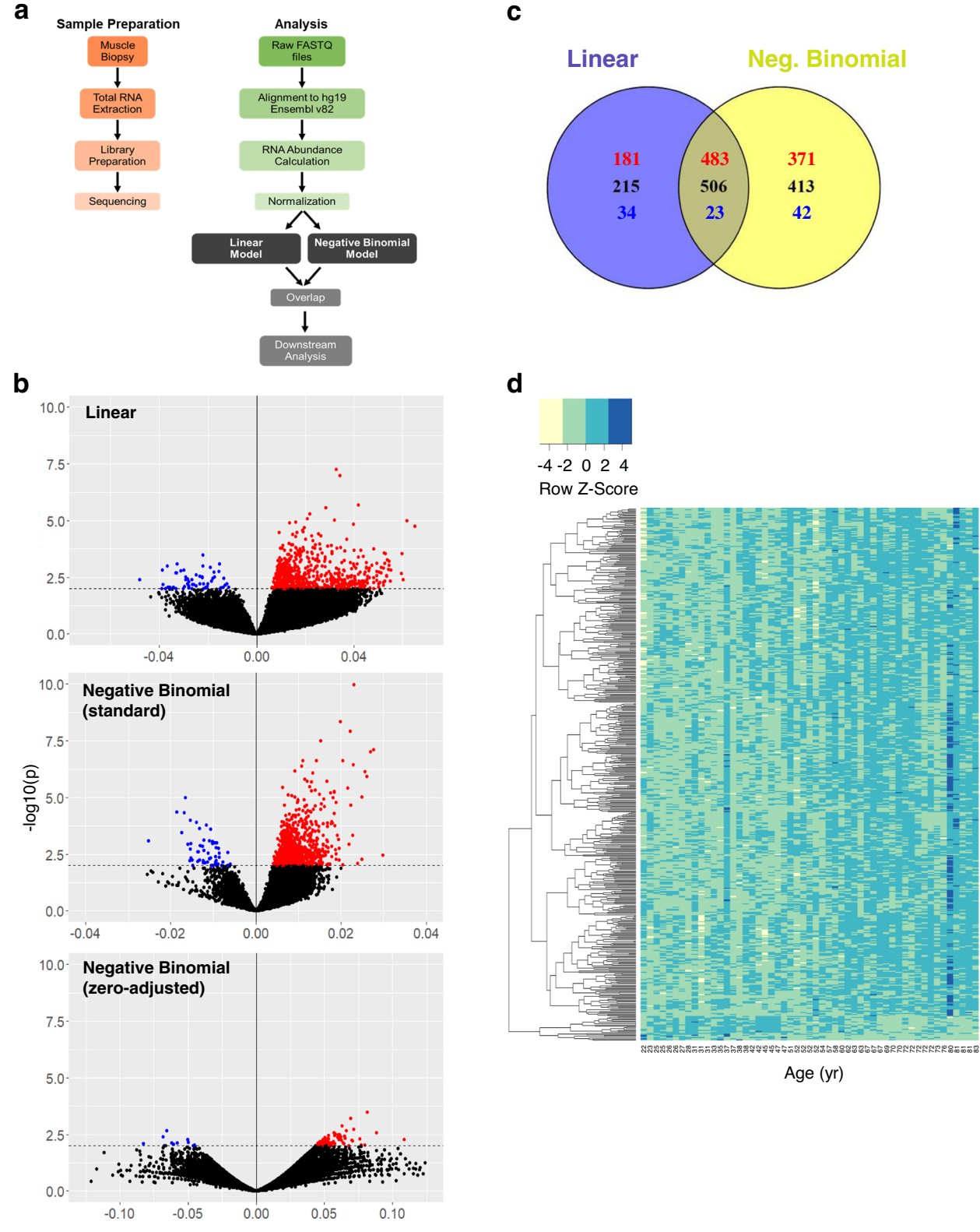

and lipid metabolism in the skeletal muscle[10,11]), *NR2F2* mRNA (nuclear receptor subfamily 2 group F member 2, important for myogenesis and skeletal muscle development[12]), and *LPP* mRNA (LIM domain containing preferred translocation partner in lipoma, implicated in smooth muscle differentiation[13]), all of which had significantly ($p < 0.01$) higher expression in older age.

Our analysis also confirmed RNA (ENSG) expression patterns with aging, previously identified in the literature, including *FAM83B* (Family with Sequence Similarity 83 Member B, involved in hypoxia response[14]), *C12orf75* (chromosome 12 Open Reading Frame 75, implicated in insulin signaling and energy metabolism[15]), *SKAP2* (Src Kinase-Associated Phosphoprotein 2, which regulates

**Fig. 1 Methodology overview, model overlap, and heat map of significant changes in RNA (ENSG) levels. a** Flowchart of sample preparation (orange) and analytical methods (green to gray) used in this study. **b** Volcano plots capture all significant ($p < 0.01$ from two-sided Wald test, unadjusted) RNAs identified by our models: linear model including all RNAs, standard negative binomial model including only RNAs with all non-zero CPB values, and zero-adjusted negative binomial model including only RNAs with at least one value of zero CPB. Red and blue points indicate significant RNAs with positive and negative $\beta$-values for age, respectively. Black points denote RNAs with nonsignificant $\beta$-values for age. **c** Venn diagram displaying the overlap between all RNAs significantly associated with aging identified by the linear (blue) and negative binomial (yellow) models (overlap shown in brown). Black denotes total significant RNAs, whereas red and blue values represent significant RNAs with positive and negative $\beta$-values for age, respectively. **d** Heat map representing the expression levels of the 506 significant RNAs shared between the linear and negative binomial models with age. Row-wise $z$-scores were calculated and color-coded to display changes in RNA expression with increasing age (22–83 years old). Light yellow and dark blue bars indicate RNAs with very low and very high expression levels, respectively, at a particular age. In addition, a correlation-based distance method was used to cluster the RNAs obtained from the linear model with positive and negative $\beta$-values for age. RNAs with positive $\beta$-values are shown in the top of the heat map, containing RNAs with low to high expression (yellow to blue rows) with increasing age. Similarly, RNAs with negative $\beta$-values are shown on the bottom of the heat map, containing RNAs with high to low expression (blue to yellow rows) with increasing age.

sarcomere function[16]), *CRIM1* (Cysteine-Rich Transmembrane Bone Morphogenetic Protein Regulator 1, implicated in smooth muscle contractility[17]), *CFAP61* (Cilia and Flagella-Associated Protein 61, highly abundant in old skeletal muscle[18]), *FEZ2* (Fasciculation and Elongation Protein Zeta 2, participating in fasciculation and axonal elongation[6]), *LGI1* (Leucine-Rich Glioma-Inactivated 1, differentially expressed in frail compared to healthy individuals[18]), and *MYLK4* (Myosin Light Chain Kinase Family Member 4, necessary for contraction, motility, and cell growth[19–21]) mRNAs (Figs. 2a, b and 3a, b, and Supplementary Figs. 2a, b and 3a,b). These mRNAs were found to be associated with skeletal muscle aging (including *C12orf75* and *LGI1* mRNAs) and with exercise (including *C12orf75* and *CFAP61* mRNAs) in previous studies performed in different populations, and using different technologies[6,9,22,23].

Linear regression plots for the RNAs (ENSGs) ranked 11–20 with positive and negative $\beta$-values for age (Supplementary Fig. 4), the top 20 RNAs identified by the negative binomial model with positive and negative $\beta$-values for age (Supplementary Figs. 2a and 3a), and negative binomial regression plots for the top 20 RNAs with positive and negative $\beta$-values for age are included (Supplementary Figs. 2b, 3b, and 5). Among the top 20 overrepresented RNAs identified by the linear and negative binomial models (Fig. 2a and Supplementary Fig. 2a), there was an overlap of 13 RNAs, of which 12 were protein-coding RNAs. Similarly, among the top 20 underrepresented RNAs obtained by both models (Fig. 3a and Supplementary Fig. 3a), there were 15 RNAs shared, of which 12 were protein-coding RNAs. Overall, 39 distinct protein-coding RNAs were identified among our four top 20 lists (Supplementary Fig. 6).

To validate our most significant results, we used reverse transcription followed by real-time, quantitative PCR (RT-qPCR) analysis. Validation was limited to a few individuals at the two ends of the age distribution (20–34 years vs. 65–79 years) for whom muscle RNA was still available. Eight RNAs that were highly associated with aging in our study, as well as in previous literature, were selected for validation. Four of the eight RNAs (ENSGs) selected for validation [*FAM83B* ($p = 0.0472$), *LGI1* ($p = 0.0283$), *NR2F2* ($p = 0.0278$), and *C12orf75* ($p = 0.0163$) mRNAs] had significantly higher expression in the old (80+ years) compared to the young (22–34 years) (Fig. 4 and Supplementary Table 1). Expression levels of *CDKN2B*, *CRIM1*, *IRS2*, and *SKAP2* mRNAs were also higher in the older age group, although not significantly, possibly due to the limited sample size ($n = 5$).

**Proteomic and pathway analyses.** Next, we considered whether differential abundance of these RNAs (ENSGs) in older skeletal muscle was reflected in differences at the protein level. For this

analysis, we selected mRNAs for which we had proteomic data, obtained from the same skeletal muscle using a shotgun liquid chromatography-mass spectrometry method[1] that yielded 4281 proteins. In agreement with previous reports, we found that the correlation coefficients of mRNAs and their corresponding expressed proteins showed a symmetrical, quasi-normal distribution around approximately zero, with more values above than below zero[24] (Fig. 5a). Similarly, adjusted regression models were used to estimate the $\beta$-coefficients for age for the 4281 mRNAs, for which corresponding proteomic data were available (age-protein-$\beta$ vs. age-mRNA-$\beta$; Fig. 5b gray dots), and separately for 122 mRNAs whose $\beta$-coefficients showed significant association ($p < 0.01$) with age (Fig. 5b red dots). Interestingly, there was a significant positive correlation between the $\beta$-coefficients, although with substantial noise. Then, the abundance of 122 mRNAs and their respective proteins was explored in the heat map analysis shown in Fig. 5c, with the correlation distribution in Supplementary Fig. 7a. As a comparison, we repeated these analyses for a randomly selected sample of 122 mRNAs. This analysis showed far less correlation with their corresponding proteins (Supplementary Fig. 7b).

By Kyoto Encyclopedia of Genes and Genomes (KEGG) annotation, the top positively correlated protein–mRNA pairs were found involved in glycolysis/gluconeogenesis (e.g., PGK1, TPI1 (Triose-phosphate Isomerase 1), PFKFB2 (6-Phosphofructo-2-Kinase/Fructose-2,6-Biphosphatase 2), and MYOZ2) and metabolism (e.g., PGK1 and LDHA (Lactate Dehydrogenase A)). Sarcomere proteins such as ANK3, PDLIM, and TNNT3 (troponin T3, fast skeletal type) also showed high protein–mRNA correlations, whereas Histone protein H2A.Z showed the strongest negative correlation (Fig. 5c inset). By functional annotation, 16 proteins were annotated as a muscle function, of which 15 were positively correlated with their respective mRNA levels (Supplementary Fig. 7c). Finally, quantitative proteomic data were available for 14 out of the top 39 mRNAs (ENSGs) showing differential expression with age; among these, four were overrepresented and 10 were underrepresented with age (Supplementary Fig. 6). Of the 14 mRNAs (ENSGs), 10 displayed significant ($p < 0.05$) correlations between protein abundance and age (Supplementary Fig. 8). Nine of these ten proteins displayed significant ($p < 0.05$) correlations with mRNA abundance. Four of these nine proteins are involved in carbohydrate metabolism, namely OXCT1 (3-Oxoacid CoA-Transferase 1, implicated in energy production from ketone bodies[25]), LDHA (involved in anaerobic glycolysis[26]), PFKFB2 (encoding a heart isozyme[27]), and TPI1 (which catalyzes the isomerization of glyceraldehyde 3-phosphate and dihydroxyacetone phosphate[28]). Three of the nine proteins are also direct constituents of the contractile system, namely MYL1 (Myosin Light Chain 1, expressed in fast skeletal muscle and associated with myopathy[29]), TNNT3 (which influences muscle contractility and

**a**

| | Rank | RNA Ensembl ID (common name) | P-value | Beta | Associated Functions & Characteristics |
|---|---|---|---|---|---|
| Protein-Coding | 1 | ENSG00000147883 (CDKN2B) | 5.55E-08 | 0.0328 | Aging[9], p15 tumor suppressor[22], aneurysm formation[63] |
| | 2 | ENSG00000168143 (FAM83B) | 9.92E-08 | 0.0343 | Hypoxia response pathway[14], aging, skeletal muscle weakness[18] |
| | 3 | ENSG00000235162 (C12orf75) | 1.99E-06 | 0.0419 | Energy metabolism, insulin signaling[15], aging, skeletal muscle weakness[18] |
| | 4 | ENSG00000108231 (LGI1) | 2.63E-06 | 0.0284 | Aging, skeletal muscle weakness[18], limbic encephalitis[84], epilepsy[85] |
| | 5 | ENSG00000005020 (SKAP2) | 5.01E-06 | 0.0218 | Sarcomere function and regulation, higher expression in older subjects[16] |
| | 6 | ENSG00000150938 (CRIM1) | 8.4E-06 | 0.0208 | Skeletal muscle aging[16], smooth muscle contractility[17], CNS development[86] |
| | 7 | ENSG00000089101 (CFAP61) | 9.63E-06 | 0.0320 | High expression in old skeletal muscle, aging, skeletal muscle weakness[18] |
| | 8 | ENSG00000185551 (NR2F2) | 1.15E-05 | 0.0160 | Myogenesis, skeletal muscle development[12], aging[23], muscular dystrophy[87] |
| | 9 | ENSG00000145012 (LPP) | 1.29E-05 | 0.0135 | Smooth muscle differentiation[13], cell migration and proliferation[88] |
| | 11 | ENSG00000196482 (ESRRG) | 2.09E-05 | 0.0196 | Type I muscle fiber development, estrogen signaling[34], obesity susceptibility[35] |
| | 12 | ENSG00000185950 (IRS2) | 2.71E-05 | 0.0189 | Insulin signaling[10], lipid metabolism in skeletal muscle[11], longevity in mice[89] |
| | 13 | ENSG00000171055 (FEZ2) | 3.07E-05 | 0.0190 | Fasciculation, axonal elongation, skeletal muscle aging[6] |
| | 14 | ENSG00000168769 (TET2) | 3.3E-05 | 0.0092 | Smooth muscle plasticity[90], skeletal myogenesis[91], DNA demethylation[92] |
| | 15 | ENSG00000185760 (KCNQ5) | 3.35E-05 | 0.0165 | Potassium channel subunit[93] |
| | 16 | ENSG00000115380 (EFEMP1) | 3.74E-05 | 0.0254 | Early aging in mice[70], lymph node metastasis, angiogenesis[94] |
| | 17 | ENSG00000132464 (ENAM) | 4.03E-05 | 0.0147 | Tooth enamel formation[95] |
| | 18 | ENSG00000181690 (PLAG1) | 4.6E-05 | 0.0263 | Higher expression in older subjects[16], bovine fetal development and body size[39] |
| | 19 | ENSG00000123836 (PFKFB2) | 4.74E-05 | 0.0225 | Heart isozyme[27] |
| | 20 | ENSG00000138439 (FAM117B) | 5.10E-05 | 0.0161 | Sarcoidosis susceptibility[96], candidate for early onset myocardial infarction[97] |
| Non-Coding | 10 | ENSG00000264151 | 1.43E-05 | 0.0397 | lincRNA |

**b**

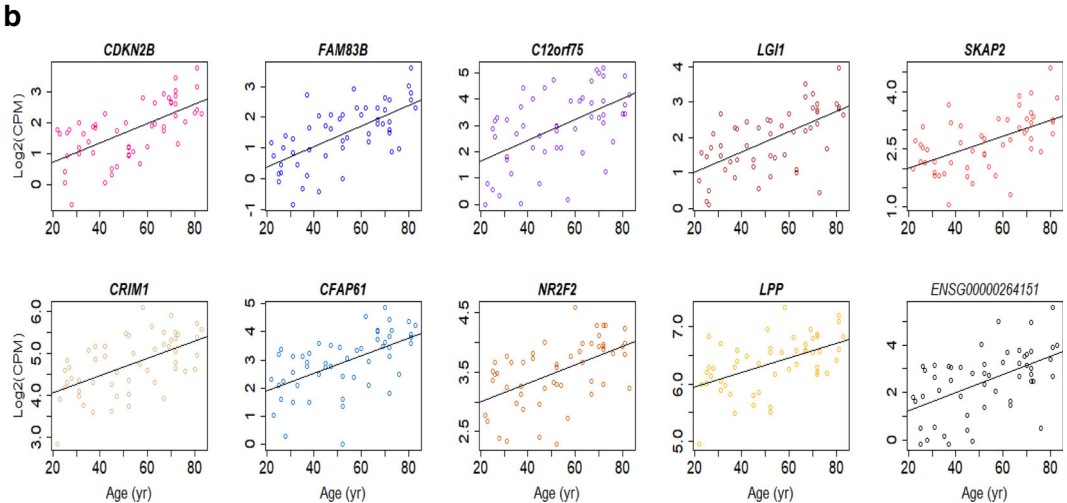

**Fig. 2 Top 20 significant (*p* < 0.01) RNAs (ENSGs) identified by linear regression with positive *β*-values for age. a** Linear model *p*-values (two-sided Wald test, unadjusted) and *β*-values, and RNA characteristics obtained from literature review, are shown. RNAs that failed to have at least ten samples with CPM values above −3 were removed. **b** Linear plots representing log2(CPM) values with age for the top ten RNAs ranked by the lowest *p*-value (two-sided Wald test, unadjusted)[36,37,84–96].

development[30,31]), and TPM1 (Tropomyosin 1, which modulates actin–myosin interactions[32]) (Fig. 5d). Similar to previous studies, and in spite of the fact that the effect of age on mitochondrial protein representation is higher than any other class of proteins, mRNAs encoding mitochondrial proteins did not show the highest correlation with their respective proteins[33].

Of the 506 RNAs (ENSGs) significantly associated with age, 478 were found in the Ingenuity Pathway Analysis (IPA) database. IPA identified 27 biological pathways that were significantly (*p* < 0.05) associated with aging, with adipogenesis being the most dysregulated. Next, we investigated the expression patterns of the input genes annotated to this pathway. Expression levels of 12 out of the 132 genes in the adipogenesis pathway were

significantly changed with age (Supplementary Fig. 9, labeled purple).

**Differential transcript usage with age.** To further explore changes in transcripts (ENSTs) that occur in the skeletal muscle with aging, we first identified specific splice variants of skeletal muscle proteins that were differentially altered with aging. For this initial analysis, we focused on genes whose RNAs (ENSGs) were among the four top 20 lists (Figs. 2a and 3a, and Supplementary Figs. 2a and 3a) with significant association with aging. We then tested for associations between differential transcript usage (DTU, see "Methods") and age as a continuous variable using linear regression, and found changes

**a**

| | Rank | RNA Ensembl ID (common name) | P-value | Beta | Associated Functions & Characteristics |
|---|---|---|---|---|---|
| Protein-Coding | 1 | *ENSG00000134333 (LDHA)* | 0.0003 | -0.0222 | Porcine energy regulation, glycogen metabolism, and glycolysis of skeletal muscle[98] |
| | 2 | *ENSG00000197872 (FAM49A)* | 0.0008 | -0.0154 | Increased proteomic expression with age in mice[99], dementia[100] |
| | 3 | *ENSG00000140416 (TPM1)* | 0.0010 | -0.0222 | Modulates actin-myosin interactions[32] |
| | 4 | *ENSG00000111669 (TPI1)* | 0.0012 | -0.0154 | Glycolysis enzyme, energy generation for muscle cells in chickens[101] |
| | 5 | *ENSG00000170290 (SLN)* | 0.0018 | -0.0156 | Inhibits rat sarcoplasmic Ca²⁺-ATPases[102], thermogenesis[103], muscle performance[104] |
| | 6 | *ENSG00000130595 (TNNT3)* | 0.0019 | -0.0175 | Skeletal muscle contractibility[30], porcine muscle development, tropomyosin binding[31] |
| | 7 | *ENSG00000139656 (SMIM2)* | 0.0020 | -0.0331 | Potential indicator of physical activity[105] |
| | 8 | *ENSG00000145949 (MYLK4)* | 0.0024 | -0.0250 | Aging, weakness[18], contraction[19], motility[20], TGF-beta pathway[21], circadian rhythms[106] |
| | 10 | *ENSG00000183785 (TUBA8)* | 0.0035 | -0.0180 | Intracellular transport of macromolecules and cytoskeletal support[107] |
| | 13 | *ENSG00000180209 (MYLPF)* | 0.0047 | -0.0185 | Fast and slow skeletal muscle development[29], lower in old caprine skeletal muscle[108] |
| | 14 | *ENSG00000173436 (MINOS1)* | 0.0060 | -0.0123 | Mitochondrial function and cristae organization[109] |
| | 18 | *ENSG00000166343 (MSS51)* | 0.0071 | -0.0131 | Skeletal muscle-specific modulator of cellular metabolism[110] |
| | 20 | *ENSG00000168530 (MYL1)* | 0.0088 | -0.0115 | Expressed in fast-type muscle[14] |
| Non-Coding | 9 | *ENSG00000257542* | 0.0031 | -0.0173 | Olfactory receptor pseudogene |
| | 11 | *ENSG00000270136* | 0.0036 | -0.0136 | MINOS1 readthrough gene (nonsense-mediated decay) |
| | 12 | *ENSG00000232407* | 0.0039 | -0.0482 | Peptidylprolyl isomerase A pseudogene |
| | 15 | *ENSG00000227258* | 0.0061 | -0.0233 | SMIM2 antisense RNA |
| | 16 | *ENSG00000184844* | 0.0062 | -0.0285 | Cytochrome c, somatic pseudogene |
| | 17 | *ENSG00000234281* | 0.0062 | -0.0202 | LANCL1 antisense RNA |
| | 19 | *ENSG00000256364* | 0.0084 | -0.0139 | MLEC antisense RNA |

**b**

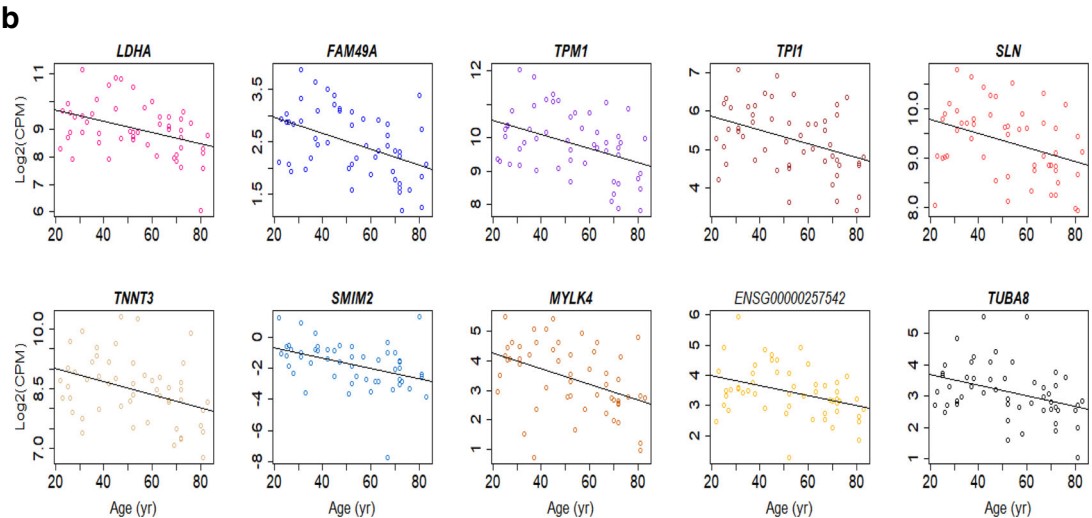

**Fig. 3 Top 20 significant ($p < 0.01$) RNAs (ENSGs) identified by linear regression with negative $\beta$-values for age. a** Linear model $p$-values (two-sided Wald test, unadjusted) and $\beta$-values, and RNA characteristics obtained from literature review, are shown. RNAs that failed to have at least ten samples with CPM values above −3 were removed. **b** Linear plots representing log2(CPM) values with age for the top ten RNAs ranked by the lowest $p$-value (two-sided Wald test, unadjusted)[97-108].

in 1005 RNAs (and their ~1100 splice variants) significantly ($p < 0.01$) associated with age. For each of the identified genes, we compiled a list of splice variants already reported in the literature that were also detected in our RNA-seq analysis. Among these, we searched for evidence of variants that changed systematically with aging and found that five splice variants were identified with significant ($p < 0.01$) changes in DTU with age. These variants were transcribed from four genes: *CFAP61*[18], *ESRRG* (Estrogen-Related Receptor Gamma, important for type-1 muscle fiber development, estrogen signaling, and obesity susceptibility[34,35]), *TET2* (Tet Methylcytosine Dioxygenase 2, involved in myogenic differentiation of skeletal myoblast cells and muscle age-related decline in mice[36–38]), and *PLAG1* (Pleiomorphic Adenoma Gene 1, known to

be more highly expressed in older humans[16,39]) (Fig. 6a and Supplementary Fig. 10). Linear regression lines for both the significant and nonsignificant age associations of all identified mRNA splice variants in these four genes are plotted in Fig. 6a, b. The expression levels of three out of these five splice variants were significantly ($p < 0.01$) higher with older age (Supplementary Fig. 11). In some cases, we found dissimilar trends in DTU with age on the percentage and log2(TPM + 1) scales. For example, the DTU of a splice variant of *PLAG1* mRNA (ENST00000423799) on the percentage scale was lower with older age, as indicated by a negative $\beta$-value (Supplementary Fig. 10b), whereas the differential levels of this splice variant on the log2(TPM + 1) scale was virtually constant with older age, indicated by a near-zero slope (Supplementary Fig. 11). To extend

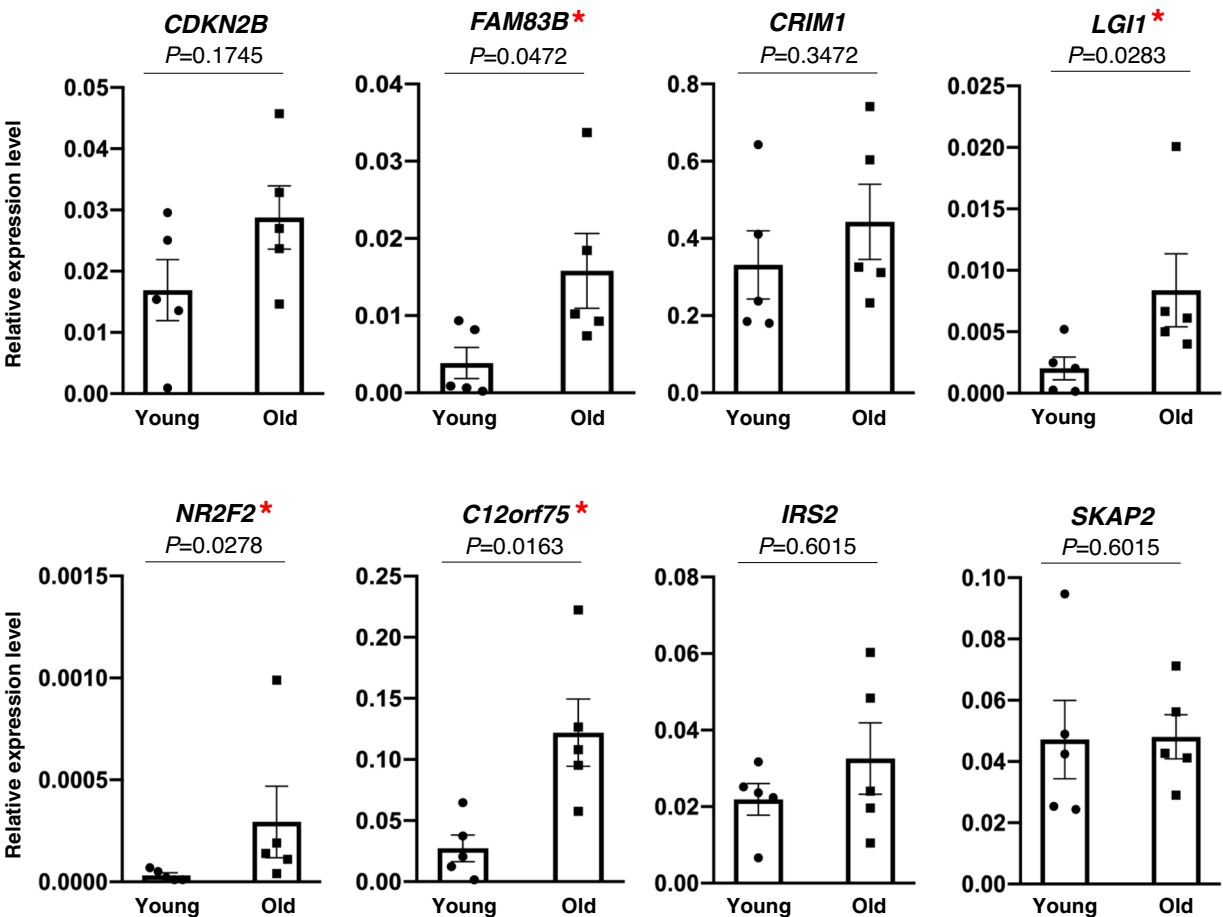

**Fig. 4 Validation of RNA (ENSG) expression patterns with age using RT-qPCR analysis.** Relative expression levels of five young (20–34 years) and five old (65–79 years) participants were calculated relative to *GAPDH* mRNA levels, and were compared using a two-sided Kruskal–Wallis test. Of the eight RNAs selected for analysis, statistically significant changes in expression with age were identified for *FAM83B* ($p = 0.0472$), *LGI1* ($p = 0.0283$), *NR2F2* ($p = 0.0278$), and *C12orf75* ($p = 0.0163$) mRNAs, denoted by a red asterisk. All *p*-values are unadjusted. Error bars represent mean ± SEM.

this analysis further, we looked at the genes that did not change significantly with age in either the linear or negative binomial model and found two transcripts that changed significantly with age (in opposite directions) from the DTU data (Supplementary Fig. 12a). Out of 890 such genes, we selected *SELENOF* (Selenoprotein F, encoding an mRNA that increases with age and is a target of microRNA mmu-mir-136, which decreases with age in skeletal muscle[40]), and *SLC47A1* (Solute Carrier Family 47 Member 1, encoding a protein that contributes to the tissue distribution and excretion of many drugs and is widely expressed in skeletal muscle[41]) for plotting (Supplementary Fig. 12b–d), selected based on absolute $\beta$-values for the transcripts. These findings suggest that the relative contribution of different splice variants to the expression of a certain gene may be as biologically meaningful as the total quantity.

**Differential exon usage with age**. As splice variants are generated by differential exon usage, we determined the level of different exons used by counting the number of reads aligned to each exon, followed by differential exon usage analysis using the DEXSeq package[42]. There were 163 RNAs (ENSGs) with statistically significant (false discovery rate (FDR) < 0.1) differences in exon usage between young (20–34 years) and old (80+ years) individuals. There were 17 RNAs that showed both significant differential exon usage between young and old participants, and significant correlation of DTU with aging (Supplementary Fig. 13a). For these RNAs, a spectral ribbon plot was created to

display changes in exon usage with age (Supplementary Fig. 13b). Next, we created transcriptome-wide UCSC browser tracks and specifically investigated these 17 RNAs. Among them, two examples were selected to illustrate age-related differential exon usage: *SMIM11A* mRNA (Small Integral Membrane Protein 11A, of unknown function) and *RXYLT1* mRNA (Ribitol Xylosyl-transferase 1, associated with Walker–Wahlburg syndrome and muscular dystrophy–dystroglycanopathy[43,44]). The UCSC browser coverage plots (read counts per million, CPM) and the age-related exonic expression levels identified by DEXSeq analysis show exons splicing in or out with age (Fig. 7 and Supplementary Fig. 14). We found that exon 11 (labeled purple in the gene model) was completely lost or significantly ($p < 0.01$) reduced in the *SMIM11A* mRNA of older individuals (80+ years) compared to younger (20–34 years) individuals (Fig. 7). Similarly, we found that exon 9 (labeled purple in the gene model) was spliced in or had significantly ($p < 0.01$) higher expression in the *RXYLT1* mRNA when comparing old to young participants (Supplementary Fig. 14). Interestingly, the significantly differentially spliced exon of *SMIM11A* mRNA (ENST00000399292) is a 3′-untranslated region with binding sites for 35 RNA-binding proteins, including ELAVL1 (ELAV-like RNA-Binding Protein 1, also known as Hu antigen R or HuR, which is highly expressed in many cancers and implicated in controlling mRNA turnover and translation), MOV10 (Moloney Leukemia Virus 10, an RNA helicase), UPF1 (Up-Frameshift Suppressor 1 Homolog, a protein involved in mRNA nuclear export and surveillance, and

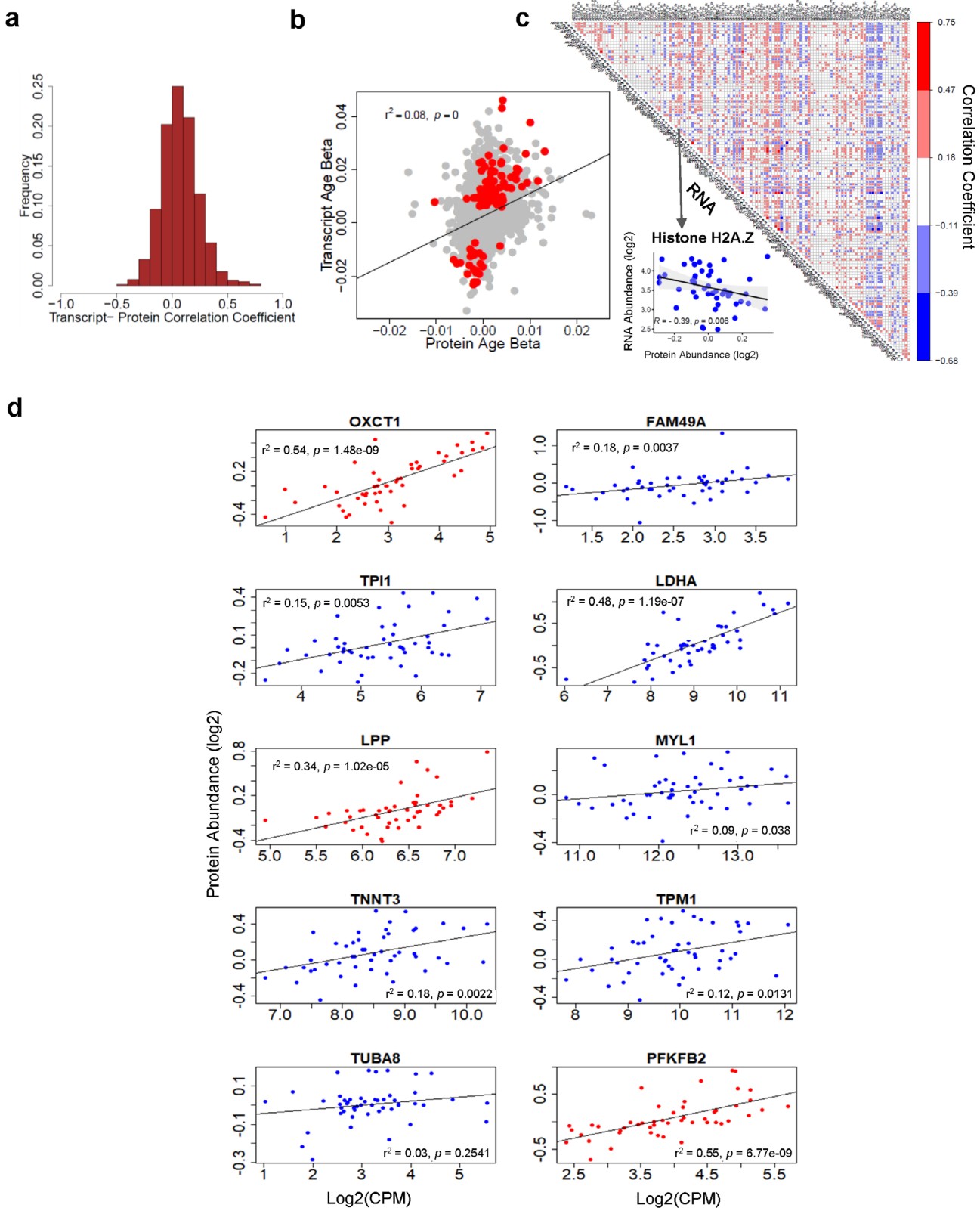

**Fig. 5 Significant (*p* < 0.05) downstream protein abundance and log2(CPM) correlations. a** Distribution of 4281 protein–RNA correlation coefficients. **b** Scatterplot of the age *β*-coefficients for the protein–RNA matches limited to the 122 mRNAs significantly associated with age (*p*-value from two-sided Wald test, unadjusted). **c** Heat map of 122 mRNAs significantly associated with age and their correlations with proteins. The mRNA with the strongest negative correlation (H2AZ) is shown in the inset (*p*-value from two-sided Wald test, unadjusted). **d** Among all the RNAs in the four top 20 lists, ten RNAs were identified to have statistically significant correlations between relative protein abundance and age (red and blue points indicate mRNAs with positive and negative correlations, respectively). Of these ten RNAs, nine had statistically significant correlations between relative protein [log2(Protein Abundance)] and RNA abundance [log2(CPM)] (*p*-values from two-sided Wald test, unadjusted).

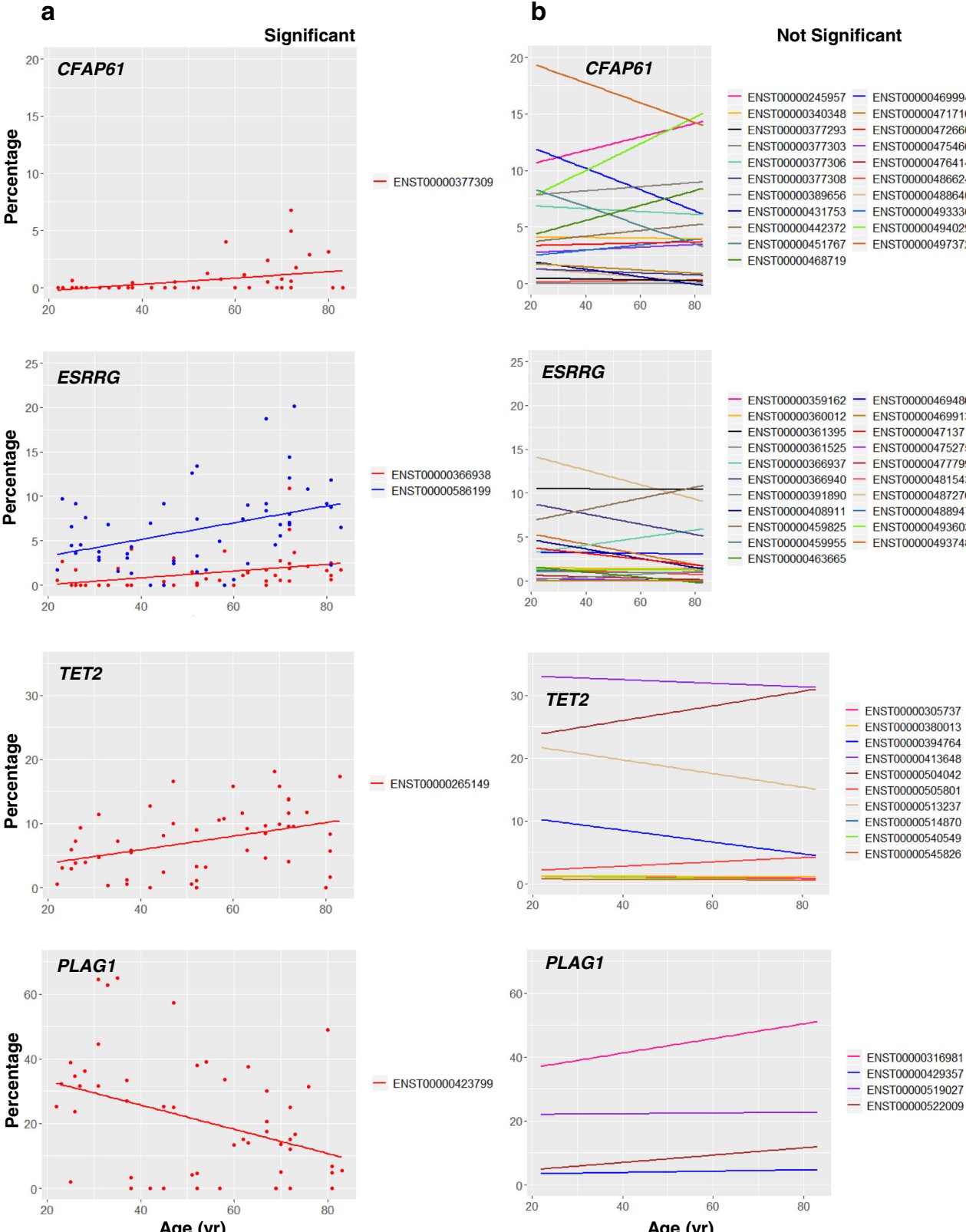

**Fig. 6 Differential transcript usage analysis for top RNAs (ENSGs).** Among the RNAs in the four top 20 lists, *CFAP61*, *ESRRG*, *TET2*, and *PLAG1* mRNAs were identified to have splice variants with statistically significant changes in differential transcript usage with age. **a** Change in differential transcript usage for all significant variants of the four mRNAs (all $p < 0.05$ from two-sided, unadjusted Wald tests). **b** Change in differential transcript usage for all nonsignificant variants of the four RNAs, provided for comparison (all $p > 0.05$ from two-sided, unadjusted Wald tests).

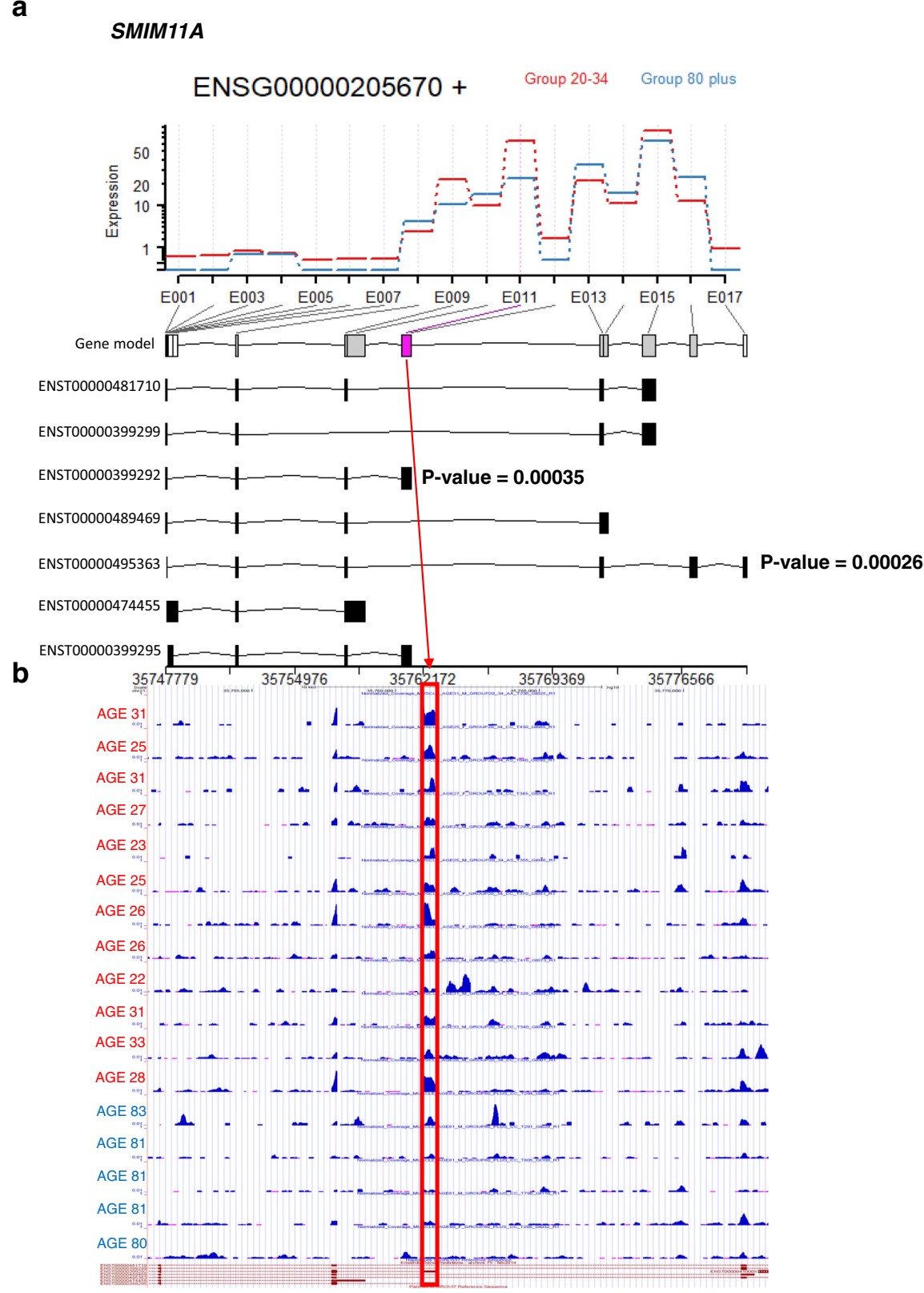

nonsense-mediated mRNA decay), and FUS (Fused in Sarcoma, a protein involved in pre-mRNA processing and mRNA export)[45,46] (Supplementary Fig. 15a). Many of these proteins bind to multiple sites along this exon and are known to regulate RNA abundance and stability (Supplementary Fig. 15b). Further analysis of *SMIM11A* and *RXYLT1* mRNAs showed that their absolute abundance did not change significantly ($p < 0.01$) with age (Supplementary Fig. 16a–d).

Gene set enrichment analysis (GSEA)[47] was conducted using the 5325 RNAs (ENSGs) with at least one significantly changing splice

**Fig. 7 Changes in the differential exonic usage of *SMIM11A* mRNA in young (20–34 years, *n* = 12) compared to old (80+ years, *n* = 5) age groups.**
**a** Red and blue bars represent the average exon usage for all 17 exons of *SMIM11A* mRNA in the young and old groups, respectively. DEXSeq analysis identified one exonic region (E011) that significantly changed in abundance between the age groups (pink in the gene model). The average exonic usage in this region is higher in the young age group and lower in the old age group. Of the seven variants of *SMIM11A* mRNA, two have statistically significant changes in expression continuously with age according to our linear model. Linear model *p*-values (two-sided Wald tests, unadjusted) are displayed for these variants. The horizontal axis below the variant models denotes the positions in the genome. **b** Ages of participants in the young (red) and old (blue) age groups are shown on the vertical axis. UCSC browser tracks indicate that the exons displaying differential expression (location on each transcript identified by red bar) identified in our DEXSeq analysis are more expressed in the young age group than in the old age group, indicated by the presence of larger blue peaks in younger participants. UCSC browser tracks support our DEXSeq findings.

variant with age. GSEA identified six significantly ($p < 0.05$) upregulated pathways in older (80+ years) compared to younger (20–34 years) participants: oxidative phosphorylation, adipogenesis, G2/M checkpoint, MTORC1 signaling, fatty acid metabolism, and ultraviolet response (Supplementary Fig. 17).

Finally, we compared the genes associated with human muscle aging from several past studies, namely Melov et al.[48], Phillips et al.[2], Sood et al.[3], and Timmons et al.[49], as well as genes induced by exercise, and we found a considerable overlap (Supplementary Fig. 18). In particular, RNAs (ENSGs) expressed from two genes were identified by all five studies including the current study (*LGI1* and *FEZ2*), 11 were identified by four out of five studies (*CAMTA1*, *COL4A5*, *KLF5*, *CYP1B1*, *USP54*, *CTH*, *PLAG1*, *NT5C2*, *TPI1*, *PFKFB2*, and *UNC13C*), 44 were identified by three out of five studies, and 228 were identified by two out of five studies. Considering all five studies, our findings were replicated for 285 transcripts, whereas 82 transcripts identified as significantly changing with age in our study were replicated in at least one other study (Supplementary Fig. 18a, b). The overlap between transcripts that changed with exercise in Phillips et al.[2] and Keller et al.[50], and those that changed with aging in our study shows eight RNAs associated with aging in our study, which were impacted by exercise in both prior studies (expressed from genes *TSPAN2*, *SKAP2*, *CAV1*, *NID1*, *JAM2*, *EDNRA*, *ABCG1*, and *GLCCI1*) and 56 RNAs impacted by exercise by at least one of these two studies (Supplementary Fig. 18c, d).

**Discussion**
In this study, we used RNA-seq data obtained from muscle biopsies collected from a group of individuals dispersed over a wide age range, who were established to be very healthy based on a strict enrollment criteria to explore transcriptomic changes that occur in healthy aging. Our aim was to ascertain transcriptional changes that occur with aging, without the potentially confounding effects of the ongoing diseases.

We found appreciable overlap between the 506 RNAs (ENSGs, containing 476 gene symbols) found in our study and prior microarray analyses that identified transcripts associated with muscle aging and exercise status (Supplementary Fig. 18). This comparison was important, as microarray analysis can be highly sensitive and quantitative, and can complement RNA-seq analysis, which offers other advantages, such as detection of previously unknown transcripts. We found that 82 age-associated transcripts identified in our study had already been linked to muscle aging in previous studies, namely *FAM83B*, *C12orf75*, *LPP*, *SKAP2*, *CRIM1*, *FEZ2*, *LGI1*, and *MYLK4* mRNAs[3,6,9,22,23,49]. In addition, eight transcripts also identified in our study were differentially regulated with frailty and were reversed by physical activity (*TSPAN2*, *SKAP2*, *CAV1*, *NID1*, *JAM2*, *EDNRA*, *ABCG1*, and *GLCCI1* mRNAs)[2,9]. The replication of these findings using different methods and in a highly selective healthy population strongly suggests that the encoded proteins are profoundly involved in skeletal muscle aging and raises the possibility that when these changes are overt, they may accompany the development of frailty.

Many of the transcripts (ENSTs) identified in our study as associated with aging encode proteins that are implicated in mechanisms of muscle maintenance. FAM83B is a member of the FAM83 protein family, characterized by the presence of a DUF1669 domain, which is implicated in a myriad of cellular processes through the regulation of casein-kinase 1, including wingless-related integration site (Wnt) signaling, which is imperative for muscle repair and maintenance[51].

C12orf75 (also known as OCC1) has been identified as a primary Wnt signaling regulator, although the specific mechanism is still unclear[52], and *C12orf75* mRNA was found to be overrepresented in the quadriceps muscles of patients with chronic pulmonary diseases compared to controls[15].

LPP is mostly expressed in the cytoskeleton of striated muscle (heart and skeletal muscle) and contributes to mechanoreception, sensing of cross-binding activity, and mitochondrial signaling. Myocyte contraction regulates the production of muscle LIM proteins (MLPs, including LPP) and their nuclear import. In the nucleus, MLPs activate multiple myogenic transcription factors, such as myoblast determination protein 1 and myogenin. In the cytoplasm, MLPs regulate autophagy, modulate the assembly of macromolecular complexes along sarcomeres and the cytoskeleton, and crosslink actin filaments into bundles[53]. Through these mechanisms, MLPs contribute to muscle mass maintenance and continuous repair. Consistent with these findings, elevated MLP levels were reported in skeletal myopathies, such as facioscapulohumeral muscular dystrophy, nemaline myopathy, and limb girdle muscular dystrophy. The specific cause of LPP dysregulation with aging is unknown, but it may reflect the need to upregulate repair mechanisms in the muscle that has accumulated damage during the aging process.

SKAP2 modulates immunity[54], influences sarcomere function and regulation, and was found to be elevated in muscle cells of older human subjects[16]. Upregulation of SKAP2 was identified in previous studies of muscle aging, both in vivo and in vitro. Microdamage likely accumulates with aging in myofibers and triggers repair by overexpressing integrins at the site of the insult[55,56]. Integrins signal for macrophage adhesion, migration, and chemotaxis, and stimulate the remodeling of the macrophage cytoskeleton, thereby contributing to changes in cell shape, motility, and directionality[57,58]. SKAP2 modulates macrophage responses to integrin signals and affects the efficiency of repair mechanisms. Therefore, overexpression of SKAP2 with aging may be a marker of active repair in response to damage accumulation. Of note, skeletal muscle *SKAP2* mRNA was recently found among a handful of transcripts that monitor a midlife metabolic switch in humans unrelated to the Target of Rapamycin complex pathway[49].

CRIM1 is essential for embryonic heart and endothelial cell development and homeostasis in the coronary vasculature, but its role in skeletal muscle physiology and aging is unknown[59]. FEZ2 acts as a cargo transport adaptor in kinesin-mediated movement and, through this mechanism, has important roles in axonal bundling and elongation, autophagy, and apoptosis[6,60]. These

activities are potentially important to support continuous rein-nervation in response to denervation that is essential for preserving muscle homeostasis with aging. The rise in FEZ2 levels with age in skeletal muscle suggests an attempt to counteract the loss of motor units and supports the notion that FEZ2 plays a role in age-related sarcopenia[16]. Interestingly, elevations in physical activity and testosterone treatment reduce FEZ2 levels, resulting in an upregulation of IGF-1 and increased muscle anabolism[61]. MYLK4 phosphorylates the regulatory light chain of myosin and plays a vital role in muscle development; polymorphisms of this gene have been associated with increased muscularity in cows[62]. Given that knowledge of the functions of these proteins is limited, especially in the skeletal muscle and aging, their deeper investigation is essential to elucidate the causes leading to age-related sarcopenia.

Our study also confirmed several age-associated mRNAs (ENSGs) encoding proteins with predicted roles in muscle aging reported by Phillips et al.[2], including CDKN2B, IRS2, and CRIM1 mRNAs. The overrepresentation of the CDKN2B mRNA with aging is particularly interesting, as the encoded protein, also known as the p15 tumor suppressor[22], inhibits cell proliferation, has been associated with cellular senescence[23], is a major hallmark of aging, and has been implicated in p53-dependent smooth muscle cell apoptosis[63]. The traditional marker of cellular senescence, p16 (CDKN2A), was not identified in the human muscle, although it increases relatively frequently in intermuscular and intramuscular adipose tissue[64].

IRS2 is strongly associated with insulin resistance to glucose uptake and lipid metabolism in the skeletal muscle, as well as type 2 diabetes[11,65], but is also involved in skeletal muscle growth and metabolism via its downstream effector molecules Akt and AMPK[66]. IRS2 mRNA expression levels decline in conditions of insulin resistance and, at least in a mouse model, account for the high risk of myocardial dysfunction associated with chronic insulin resistance and diabetes[67]. Whether decline in IRS2 also negatively affects muscle function, independent of its substrate uptake functions, is unknown. Although there is abundant epidemiological and experimental evidence that insulin resistance increases with aging, humans over the age of 90 years have less insulin resistance than might be expected and centenarians are surprisingly insulin-sensitive, suggesting that healthy aging and extreme longevity are strongly connected with the avoidance of glucose intolerance and diabetes[68]. The consistently high levels of IRS2 transcripts in older compared to younger participants may be due to the fact that the strict selection criteria for "very healthy" used in this study selected older individuals that had escaped major chronic diseases, including cardiovascular pathology, cancer, and diabetes, and therefore did not become glucose intolerant. High IRS2 mRNA levels in the older age group of these healthy and highly functional individuals is also consistent with the hypothesis that compromised insulin signaling plays a role in the development of sarcopenia and frailty[66,69].

CRIM1 has been implicated in pelvic smooth muscle contractility in mice[17] and skeletal muscle aging in humans, where its expression levels were elevated[6]. Other examples of mRNAs with higher expression in older age include PLAG1 mRNA[16,39] and EFEMP1 mRNA (EGF-Containing Fibulin Extracellular Matrix Protein 1, which induces early aging in mice[70]). Based on the strong connection of these known genes to muscle physiology, we propose that all 506 significant (p < 0.01) RNAs obtained by both the linear and negative binomial models appear robustly involved in healthy skeletal muscle aging.

IPA was used to identify specific biological mechanisms associated with the 506 significant (p < 0.01) RNAs (ENSGs) found by both models. Similar to Phillips et al.[2], we identified several significantly enriched pathways linked to the top 506 differentially abundant RNAs. The most significant pathway associated with this subset of genes, adipogenesis, did not show directionality of regulation with age. However, it is widely known that aging is associated with increased adiposity within the muscle tissue. Of interest, some of our top mRNAs implicated in adipogenesis, including SOX9 mRNA (SRY-Box 9, necessary for avoiding skeletal malformation) and SIRT1 mRNA (Sirtuin 1, which influences skeletal muscle differentiation, cellular senescence, and aging), encode proteins with essential roles in skeletal development, inflammation, cellular senescence, and aging[45].

After analyzing changes in total RNA, further analyses were aimed at understanding whether specific mRNA splice variants among the RNAs (ENSGs) in the 4 top 20 lists (Figs. 2a and 3a, and Supplementary Figs. 2a and 3a) were significantly associated with aging. In particular, we aimed at understanding whether the relative contribution of different splice variants in a specific gene changed systematically with aging.

DEXSeq and linear regression analyses identified SMIM11A and RXYLT1 mRNAs as having significant (p < 0.01) changes in exon usage and splice variant usage with age. The biological underpinnings of these age-related changes are unclear, but changes in alternative splicing with aging have already been suggested from proteomic studies showing that skeletal muscle proteins that belong to the splicing machinery are massively overexpressed with aging and significantly underexpressed in response to physical activity[1,71]. Consistent with these results, we found that the group of genes identified as having at least one splice variant changing with aging was enriched for proteins related to oxidative phosphorylation, adipogenesis, and MTORC1 signaling. Based on these findings, we propose that alternative splicing is an adaptive mechanism aimed at counteracting the loss of efficiency of oxidative phosphorylation that is often observed with aging and restored by physical activity. This hypothesis should be explored in future studies.

Our study has a number of features that provide valuable contributions to the literature. We cataloged muscle transcriptomes from a human population dispersed over a wide age range, who were confirmed to be healthy based on very strict inclusion criteria. We used state-of-the-art RNA-seq technology that has not been previously used to study skeletal muscle in a human population of this size. To identify RNAs that systematically change with aging in healthy skeletal muscle, we performed statistical analyses using both linear and negative binomial regression models, and integrated both analyses. Thus, we could identify transcripts whose abundance change homogeneously or exponentially with aging. It is likely that other meaningful age-related patterns of transcript change exist that could not be detected by our analysis (e.g., early- or late-stage changes). However, detecting these changes with a sufficient degree of confidence would have required a larger sample size and this should be addressed in future studies. Of note, the two statistical models used in this study identified considerably more RNAs with positive $\beta$-values than negative $\beta$-values for age, which is commonly observed when analyzing proteomic and transcriptomic data. We validated eight highly significant RNAs identified in our study by performing RT-qPCR analysis for a number of key transcripts and confirmed the directionality of the changes observed by RNA-seq and proteomic analyses. Unfortunately, we did not have enough tissue samples to perform a more extensive validation.

In conclusion, our study replicated the findings of multiple prior studies performed in different populations and with different technologies (Supplementary Fig. 18), but also found other transcripts for genes whose implications in the aging process have not been previously investigated. We found evidence that some specific splice variants change systematically with aging, and that

this age-associated trend is not necessarily limited to mRNAs that change overall with aging. We note that changes in splice variants with aging tend to cluster in proteins that are involved in oxidative phosphorylation, lipid metabolism, and MTORC1 regulation, raising the possibility that alternative splicing of these genes may be compensatory to the decline of mitochondrial function with aging. These findings are particularly meaningful, because they are devoid of the potentially confounding effects of disease, as we studied a larger group of very healthy, older individuals. However, recruiting such a study population is challenging and some limitations intrinsic to the limited sample size, such as nonlinear changes in the expression and a more comprehensive identification of age-related splicing events, should be considered in future, larger studies. Elucidating the mechanisms that link these transcripts to aging will provide valuable insights into the pathophysiology of age-related sarcopenia and will uncover targets for preventive and therapeutic interventions.

## Methods

**Study population.** Skeletal muscle biopsies were obtained from very healthy GESTALT participants using the methods outlined in Ubaida-Mohien et al.[1]. Participants were excluded from the study if they consumed medication for chronic illness, trained professionally, had a body mass index $\geq 30$ kg/m$^2$, or were diagnosed with cognitive damage, a physical impairment, or a major disease. The Clinical Research Unit of the National Institute on Aging developed these criteria based on medical history, physical exams, and blood tests evaluated by a trained nurse practitioner[72]. Muscle RNA-seq data obtained from the left vastus lateralis of 53 participants over a large age range (22–52 years, $n = 28$; 53–83 years, $n = 25$) was used for this study. All participants underwent a consenting process that entailed a detailed description of the study, including potential risks. All participants signed an informed consent (provided in Supplementary Information) to participate in the study. The study protocol was approved by the NIH Institutional Review Board (provided in Supplementary Information) and complied with all ethical regulations for research with human subjects.

**Sample preparation and sequencing.** After RNA was extracted from the skeletal muscle biopsies, cDNA libraries were prepared and analyzed (Fig. 1a). Total RNA was isolated in QIAcube, Qiagen, cDNA was synthesized using the NuGen Ovation v2 system, and Illumina libraries were generated with the TruSeq ChIP Library Preparation Kit (Sets A [IP-202-1012] and B [IP-202-1024]). RNA was sequenced using the Illumina HiSeq 2500 sequencing system at a depth of > 80 million single-end reads. Out of the 57,773 RNAs in the Ensembl hg19 v82 (September 2015) database, we identified, on average, 24,453 (ranging from 16,203 to 36,119) RNAs at a coverage depth of $\geq 10$ reads.

**Alignment and quality control.** After sequencing, output data in the form of FASTQ files (combination of FASTA sequences with corresponding quality data) was cleaned using cutadapt (v2.7)[73]. Quality was checked using the fastqc program (v0.11.8)[74]. Reads were aligned using STAR Aligner (v2.4.0j), a high-throughput alignment software capable of mapping full-length RNA sequences and discovering non-canonical splices and fusion transcripts[75]. BAM files obtained as the output from the STAR alignment were then processed using featureCounts (v1.4.6-p5) from the Subread package (v1.6.4)[76], to count reads to various genomic features such as exons, transcripts (splice variants), and genes (composite model of all splice variants). Further data filtering and statistical analyses were conducted using RStudio (version 1.2.1335) and multiple R libraries.

**Regression models and visualization.** All RNAs that were not expressed across any of the participants (total 568) were removed from the regression analysis (57,773–568 = 57,205 remaining). To analyze RNA (ENSG) expression levels continuously with age, read counts for 57,205 RNAs were converted to log2-transformed CPM using the edgeR package (multiple versions – 3.22.5)[77]. This log2(CPM) transformation was implemented for normalization. Next, linear regression models were built for each RNA using the MASS package (v7.3-51.5)[78]. For each RNA, we also built a negative binomial regression model, a method commonly used to analyze count data (non-negative integer values). Moreover, given that RNA-seq data mostly follow a negative binomial distribution[79], we employed an alternate strategy, as the negative binomial distribution requires input of integer values $\geq 0$. To address this obstacle, all read counts were instead converted to normalized counts per billion (CPB)[80] and then rounded to the nearest integer for negative binomial regression using glm.nb from the MASS package. Rounding was completed after converting to CPB, rather than CPM, to minimize noise. As most CPM values were only single- or double-digit integers, rounding these small numbers would have introduced much larger error. In addition, we

incorporated a zero-adjustment in the negative binomial models for RNAs with one or more values of zero CPB using hurdle from the pscl package (v1.5.2)[81]. Standard zero-inflation was not used, as our data included only structural (true) zeros and no sampling (random) zeros. Both the linear and negative binomial models included only age and sex as covariates. Of the 57,205 RNAs, there were 544 RNAs with excess zero counts removed from the negative binomial model, in which 203 of these RNAs contained CPB values of zero in > 50 samples. Over-abundance of zeros prevented model construction. An additional 13 RNAs were removed due to very low expression levels (all median CPB < 630). P-values and $\beta$-values from both models were extracted, sorted, and compared to identify RNAs with the most significant changes in expression with age. Only RNAs with at least ten log2(CPM) values of $\geq -3$ were included in our linear model results. Similarly, only RNAs with at least ten CPB values of $\geq 125$ were included in our negative binomial model results. Both of these thresholds translate to 10 reads of RNA per 80 million total aligned reads for an individual and were implemented to combat very low expression levels and noise.

Next, volcano plots were created to visualize the distribution of RNAs (ENSGs) with statistically significant ($p < 0.01$) expression patterns with age. A conservative $p$-value threshold of 0.01 was used to obtain only RNAs with the most significant changes in expression with age. Volcano plots were generated for the linear and negative binomial model results, in which red and blue points indicate significant RNAs with positive and negative $\beta$-values for age, respectively. Two separate volcano plots were created to display the standard and zero-adjusted negative binomial model results. The shared RNAs between the linear and negative binomial models were used for further analysis. A heat map was created using heatmap.2 from the gplots package (v3.0.1.1)[82] to visualize the expression level changes of these significant ($p < 0.01$) RNAs using the standard complete linkage (hierarchical) clustering method. Row-wise $z$-scores were calculated for each RNA and color-coded based on the expression level across age, in which yellow and blue bars represent RNAs with low and high expression, respectively, at a particular age.

The top 20 significant ($p < 0.01$) RNAs (ENSGs) shared by both models with positive and negative $\beta$-values for age were studied at a greater depth. We explored the functions and characteristics of these top RNAs through literature review, to identify musculoskeletal processes and other related biological mechanisms found to be associated with these RNAs in previous major studies. GeneCards (v4.12)[45] was also used to explore RNA and protein functionality. Linear and negative binomial plots were generated to display the expression patterns of these highly significant ($p < 0.01$) RNAs as a function of aging. In addition, median read CPB within the five age groups were calculated and included in each negative binomial plot, denoted by red asterisks.

**Validation by RT followed by real-time qPCR analysis.** We validated a subset of the top 20 significantly changed RNAs identified by both regression models using RT-qPCR analysis. These eight highly significant ($p < 0.01$) RNAs obtained from the linear model with positive $\beta$-values for age were selected based on literature review. For each RNA, relative expression levels of five young participants (20–34 years) were compared to five old participants (65–79 years) using the Kruskal–Wallis test. Relative expression levels were normalized to levels of the GAPDH mRNA, encoding the housekeeping protein GAPDH. We used a non-parametric test, because our limited sample size ($n = 5$) in each age group prevented us from assuming relative expression levels originated from the same underlying population. We were constrained to a small sample size and limited amounts of RNA sample obtained from each muscle biopsy. Therefore, we were unable to expand the RT-qPCR validation in this study. We also recognize that using GAPDH mRNA, which is commonly employed for normalization, is a limitation of this study due to its role in glycolysis, an age-modulated metabolic pathway. This shortcoming may account for our low validation rate. A complete list of primers used for RT-qPCR, including names and sequences, is provided in Supplementary Information.

**Proteomic analysis.** Using the 4281 proteins reported by Ubaida-Mohien et al.[1], we performed correlation analysis with the corresponding mRNAs (ENSGs). In further analysis, 122 age-associated mRNAs were correlated with their respective proteins and correlation coefficients were measured. A heat map of correlations between these mRNAs and proteins was generated using the Corrplot (v0.84) library in R (v3.6.1). A random set of 122 mRNAs and their corresponding 122 proteins were chosen for random analysis. We then selected the four top 20 transcript lists to identify RNAs with statistically significant ($p < 0.05$) correlations between relative protein abundance and age. Correlations between relative protein abundance and log2(CPM) helped us to evaluate relationships between transcriptional and translational changes in expression. This also allowed us to emphasize the biological importance of the RNAs changing most significantly with age.

**Pathway analysis.** We conducted IPA (summer 2019 release) to explore the biological processes associated with the significant ($p < 0.01$) RNAs (ENSGs) obtained from both regression models. For this analysis, input genes were matched to annotated pathways (e.g., the KEGG adipogenesis pathway is annotated with 132

genes). Input genes that were matched to the pathways were counted and the significance of the pathways was measured by calculating z-scores based on input RNA p-values and β-values for age from linear regression, and the activated state, a predetermined subset of the most active genes in a pathway. After correcting for technology and tissue bias, adipogenesis remained the most significantly changed pathway identified by IPA.

**Splice variant and exon usage analyses**. Next, we investigated which specific forms of the top RNAs (ENSGs) identified by both models with positive and negative β-values for age were most significantly associated with healthy skeletal muscle aging. All splice variants (ENSTs) considered were obtained from the Ensembl hg19 v82 (September 2015) database. Kallisto (v0.44) software was used to find transcripts per kilobase million (TPM) values of 196,354 splice variants, associated with the 57,205 total RNAs analyzed previously. These TPM values were converted to log2-normalized transcripts per million [log2(TPM + 1)]. Instead of assessing absolute splice variant expression patterns with age, splice variant reads for each participant were converted into a percentage of the participant's total variant counts for each top RNA. This percent change method was used to determine how DTU changed with age. Afterwards, linear regression models were built and plotted using the ggplot2 package (v3.2.1)[83] to visualize how the proportions of variant reads changed with respect to all other variants of each significant RNA (ENSG) with age. All p-values and β-values were extracted to obtain candidate splice variants (ENSTs) involved in healthy skeletal muscle aging. In this exploratory step, a less conservative p-value threshold of 0.05 was used for all splice variant analysis. In addition, for each significant splice variant identified, a linear model was constructed and plotted to display its expression in read CPM with age.

We also used the DEXSeq package (v1.26)[42] to investigate changes in differential exon usage between young (20–34 years, n = 12) and old (80+ years, n = 5) participants. This strategy allowed us to identify major skipped exons. Data were converted to bed files for DEXSeq analysis. Here, a more stringent FDR-adjusted p-value threshold of 0.1 was used to obtain the most significant candidate RNAs showing group-wise differences in exon usage among their splice variants. In parallel, we also assessed splice variant (ENST) expression patterns with age using linear regression and a p-value threshold of 0.01, and compared these findings to our DEXSeq results. A conservative p-value threshold of 0.01 was used, as correction for multiple testing (Benjamini–Hochberg) drastically reduced the number of significant transcripts, likely due to the small sample size (n = 53). Of all the shared RNAs, two RNAs with highly significantly differentially expressed splice variants were selected for further analysis. These expression levels were visualized using the UCSC Genome Browser, to determine whether age-related changes in exon expression within respective regions corresponded to the regions identified in our DEXSeq analysis.

Lastly, GSEA (MSigDB v6.2)[47] was conducted to explore the biological pathways associated with the RNAs (ENSGs) with at least one significantly changing splice variant with age, obtained from linear regression. For this analysis, we again compared young (20–34 years, n = 12) and old (80+ years, n = 5) participants.

**Reporting summary**. Further information on research design is available in the Nature Research Reporting Summary linked to this article.

## Data availability
All RNA-seq data used to generate Figs. 1–3, 6, and 7, and Supplementary Figs. 2–5 and 9–17 are deposited in GEO (GSE164471), and data obtained from the Ensembl hg19 v82 (September 2015) database can be located here: ftp://ftp.ensembl.org/pub/grch37/release-82. The mass spectrometry proteomics data used to generate Fig. 5 and Supplementary Figs. 6–8 have been deposited to the ProteomeXchange Consortium via the PRIDE partner repository with the dataset identifier PXD011967. Additional data inquiries or requests can be directed to the corresponding author [L.F.]. Source data are provided with this paper.

## Code availability
All R code used for this study is available upon request from the corresponding author [L.F.].

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

## Acknowledgements

We thank all study participants and the teams at the Laboratory of Genetics and Genomics, and the Translational Gerontology Branch of the National Institute on Aging for facilitating our work. This work was supported in its entirety by the National Institute on Aging–Intramural Research Program of the National Institutes of Health (NIH). We also thank Dr. Fred E. Indig (NIA, NIH) and Dr. Maxim Artyomov (Washington University, St. Louis), and their teams, for their contributions to this study.

## Author contributions

R.A.T. III carried out statistical analyses and manuscript preparation. A.H., G.K., C.U.M., and C.C. completed additional statistical analyses. J.H.Y., M.G.F., M.K., A.L., W.H.W. III, and Y.P. carried out laboratory investigations. L.M.Z. and C.W.C. recruited patients and collected medical data. J.D., M.G., R.S., and S.D. provided guidance for statistical analyses and biological insights, and assisted with manuscript preparation. L.F. planned and designed the study, and served as the principal investigator. All authors commented on the draft manuscript.

## Funding

## Competing interests

The authors declare no competing interests.
