## [Peer Review File · Nature Communications]

Reviewers' Comments:

Reviewer #1:

Remarks to the Author:

The authors have completed an extensive and very interesting analysis. The following comments are fully intended to enhance the clarity of the article and ensure that the data analysis is as comparable as possible with other studies in the literature.

Introduction

Clarification of technology utilised and its strengths and weaknesses

In the introduction RNA sequencing is put forward as an advantage of the present study over previously published microarray studies. This motivation should be tempered by empirical observations that unfortunately RNA-sequencing is neither as robust nor comprehensive as first thought while the sequencing consortium concluded that arrays and sequencing are complementary.

The lack of comprehensive transcriptome coverage is a reflection of the inherent limitations of sequencing, whereby the most abundant RNA species are sequenced over and over again, while the vast majority are not quantified as robustly or in a linear manner (Lindholm et al., 2014; Sood et al., 2016; Timmons et al., 2018).

Further, the preparation of the sequencing library results in up to 50% of the transcriptome being missed - while the majority of this transcriptome can be detected using the latest 'tiling' type arrays where the 'DNA' library prepared from the RNA covers a broader range of RNA species, not just RNA with long 3' poly-A tails (Lindholm et al., 2014; Sood et al., 2016; Timmons et al., 2018).

In addition to these laboratory factors, there are also additional assumptions and challenges assembling the raw data that limits data interpretation. It should be noted, that high density arrays should always be re-annotated to the latest genome build and thus the retained short probes remain current and accurate.

It is for these reasons that the authors would be more correct to refer to limitations of both methods, and older array technologies while to date - including the present study - a substantially greater proportion of the transcriptome is still captured in practice, with properly configured high-density 'tiling' type arrays (Timmons et al., 2018).

This will be true also for the splicing analysis where the authors present 4 robust splicing events in the present analysis - somewhat less than reported elsewhere. It should also be made clear in the abstract that many comparisons were based on 12 younger subjects and 5 older subjects which is insufficient to generate robust statistical analysis

Study population

A second strength of the present study over and above existing studies is that access to very healthy individuals enrolled according to strict clinical and functional criteria was possible. While this is undoubtedly correct for many of the smaller studies carrying out molecular profiles in human muscle, the present study did not present metabolic or cardiovascular capacity measures to clarify the physiological status of the subjects while several other studies of human muscle age have done so.

It may be more prudent to reflect, in the introduction, on the full range of major studies exploring muscle age in humans (combined with transcriptome measures) to better highlight the advantages

of the present study. It is note worth that the author have not cited two of the largest and most comprehensive studies, both of which relied on well defined subjects i.e. Phillips et al PLOS Genetics 2013 and Sood et al Genome Biology 2015. The former utilized a BBSRC funded cohort of healthy young. Middle-aged and older subjects that were extensively characterized.

The authors report that "knee strength (N m) increased over the first three age groups [Group 1 (20-34 yr, n=12), Group 2 (35-49 yr, n=11), and Group 3 (50-64yr, n=12)] and decreased beyond Group 3 [Group 4 (65-79 yr, n=13) and Group 5 (80+ yr, n=5)]. This is stated as being the expected trend in a healthy population but to me this seems like an unusual pattern and one that is not consistent with the accepted muscle physiology responses to age.

For example, the analysis by Lindle et al <https://www.ncbi.nlm.nih.gov/pubmed/9375323> or Kemmler et al <https://www.ncbi.nlm.nih.gov/pubmed/30443219> indicate that muscle strength is typically stable from 20y to 50yr and then declines. This suggests that the small subgroups of individuals in the present study are not necessarily representation of larger healthy aging populations and it is strongly recommended that knee extensor is utilised as a covariate in the present linear analysis (particularly given the fact that muscle function itself is robust altered by physical training, to some extent independent of age).

Results

The authors used the Illumina HiSeq 2500 sequencing system at a depth of >80 million single-end reads and detected, above a threshold of 10 reads per 80million, to identify 57,205 RNAs. It would be helpful to clarify how many unique genes this represented.

Their combined model identified 506 RNAs (again gene numbers would be useful). The authors present only the top 20 genes and it would be instructive to the readers to understand how many of the 506 RNAs were reported by 4 other major studies (Melov et al., 2007; Phillips et al., 2013; Sood et al., 2015; Timmons et al., 2019).

It would also be useful to define how many are related to exercise status by cross referencing with these two robust exercise transcriptome signatures (Keller et al., 2010; Phillips et al., 2013). Consideration of pathway level and gene-id level overlap will enable a more complete consideration of this questions. It is particularly important given the trend for middle-aged subjects to have greater muscle function (in males) in the present study.

qPCR validation.

The authors might wish to reflect on the 50% validation rate for the qPCR carried out on the top 20 regulated genes. They used GAPDH as the house keeping gene, which is a gene within an age modulated metabolic pathway (glycolysis). The authors own data indicate this pathway is modulated and this may be one reason for the low validation rate.

The other consideration is that qPCR applied to only the top 20 is not really a "validation" of global transcriptomics list, as the vast majority of the 506 will not have the same statistical significance as the top 20 (i.e. it's the best case scenario). It would be more robust to compare the 506 directly with other major studies in the field - which provide supplements with the full gene lists. Meanwhile I would not use the term 'validate' if you are referring to validation of your 506 list.

Pathway analysis

The Ingenuity Pathway analysis (and other such ontology analyses) require correction for technology (database) and tissue bias before using p-values generated by the software. IPA allows you to upload the list of 'detected' genes in your experiment to use as the background 'sampling' universe for calculating p-value enrichment values. Without this correction, the pathway analysis

p-values are not correct.

Splice variant section

Please clarify why 0.1 FDR was selected - given the problems with false positive data for splicing modeling, and if the linear regression model that reports 5325 spliced RNAs related to age at $p < 0.05$ is an unadjusted p-value? If it is, please report how many were significant after correction for multiple testing. A similar issue regarding GSEA and tissue sampling bias exists when the input list is based on an unadjusted p-value (it becomes a random sample of muscle expressed genes, which will reflect muscle biology over and above a random sample of the genome).

Discussion

A supplementary XL file with all regulated genes would greatly enhance the usefulness of the present analysis for other laboratories and should be included. Along with gene ID the precise ENST for the RNA species detected should be included.

For example, 7 (LGI1, SKAP2, CFAP61, NR2F2, FEZ2, PLAG1 and PFKFB2) from your top 20 in Figure 3 were reported by Timmons et al 2019 including as you mention SKAP2. Interestingly SKAP2 was first reported by Sood et al 2015 and this article should be cited.

The Sood classification model of muscle age has been re-explored and for muscle tissue age unquestionable re-validated in 2019 (See online supplement in the following new article - <https://genomebiology.biomedcentral.com/articles/10.1186/s13059-019-1734-z>)

Reviewer #2:

Remarks to the Author:

This is a very interesting research. The specific goal of this RNA-seq study reported here is to elucidate differences in the transcriptomic network of skeletal muscle as a function of age. I have some concerns on data analysis, especially on splice variant analysis.

1. Analysis tools and dataset

- Can you deposit this dataset in a public repository such as GEO or SRA?
- Can you provide more information and analysis metrics on your RNA-seq libraries as a supplementary table, such as RIN, library size, mapping rate and etc? What are your criteria for inclusion/exclusion of a sequenced sample on your analysis?
- For open source tools such as STAR, subread, edgeR, DEXSeq, R package MASS, and pscl, what are the exactly versions you are using? Please always specify the version numbers.
- What are your reference genome and transcriptome in quantification?
- For isoform (splice variant) quantification, what tool (and its version) is used?

2. Genes of interest

In this paper, the authors focused on only those genes consistently go up or down with age. Surely, other patterns of expression can be very interesting as well, for example, first go-up and then go-down; or first go-down and then go-up. Of course, we cannot use a simple linear model to identify genes with such patterns. Would it be possible to detect such genes?

3. Regression model

- For linear regression, $\log_2(\text{CPM})$ is used if I understand correctly. If so, can you justify the benefits of $\log_2(\text{CPM})$ over CPM?
- It's a common practice to add an offset in \log_2 transformation. Surely, the right offset is debatable. The default offset in edgeR is very small. Assume a sample has a total of 100M counted reads, and one gene (or transcript) has a single read mapped to it. Therefore, its corresponding

$\log_2(\text{CPM})$ is -6.6 ($=\log_2(0.01)$). In practice, the # of counted total reads are typically between 30M and 60m in a sample, therefore, the minimal $\log_2(\text{CPM})$ is usually >-5.6 for an expressed gene. For $\log_2(\text{CPM})$, any negative number smaller than -5.6 is not biologically meaningful. Without a reasonable offset in \log_2 transformation, a very large negative number can be generated, and this in turn will cause trouble for regression. For instance, the 2nd transcript of ESRRG in one sample (in Supp Figure 11) has a $\log_2(\text{CPM}) < -10$, far away from all other samples. I strongly suggest to add "1" as the offset in \log_2 transformation. The default offset edgeR is not ideal for regression.

- Gene expression = sum (Transcript expression). If so, I have difficulty in harmonizing the gene expression of CFAP61 in Figure 2 with its corresponding transcript expressions in Supp Fig 11.

4. Splice Variant and Exon Usage Analyses

Lior Pachter had an interesting post (<https://liorpachter.wordpress.com/tag/differential-expression/>) on different kinds of differential expression. Isoform switches can occur without gene-level differential expression. Differential transcript usage is a question sort of independent of differential transcript expression. Changes on both usage of isoforms (resulting from alternative splicing) and expression of isoforms with age are biologically interesting. May I suggest a more comprehensive analysis of this section?

a. Perform isoform-level expression analyses as you have already done at the gene level, and then compare the results. Typically, gene expression is dominated by a single or a couple of isoforms.

b. Perform transcript usage analysis, and identify genes with potential isoform switches. What are those genes and what they do? It's most likely you will get more insights from this RNA-seq dataset.

c. To my knowledge, DEXSeq can detect exon inclusion and exclusion only, but not other kinds of splicing events, such as alternative 5-end splicing site, alternative 3'-end slicing event, or alternative polyadenylation. Some tools are particularly designed for alternative splicing, including rMATs, MAJIQ, leafcutter and etc. My gut feeling is that there are a lot of interesting changes in splicing during aging. Such splicing events can be potential predictive biomarkers, and become drug targets of 'longevity'. If so, your research will impact a much broader community who are interested in ageing.

Minor typo in Supplementary text – "Supplementary Fig. 5. Proteomic analysis" should be "Supplementary Fig. 7. Proteomic analysis".

Reviewer #3:

Remarks to the Author:

The paper named: "Skeletal Muscle Transcriptome in Healthy Aging" by Tumasian III, R.A et al. provide a careful analysis of transcript levels. The study identify 57,205 protein-coding and non-coding RNAs, among them 1,134 RNAs were significantly associated with age. The cellular processes involved are: cellular senescence, insulin signaling, and myogenesis as expected. Furthermore, the careful analysis of splice variants indicate that were enriched for proteins involved in oxidative phosphorylation and adipogenesis. The study is original since a previous study published many years ago on muscle proteome analysis of healthy subjects is nowadays overcome by more efficient technical approaches even though some suggestion for interpretation of present data in the frame of muscle physiology can be of some aid (particularly for alternative splicing). Although it is trough that no previous study collected detailed information in very healthy individuals enrolled according to strict clinical and functional criteria and in a quite large number of samples with 5 individuals over 80. Nevertheless, the number of sample still represent a limitation in "omic" studies and verification adopting alternative technologies, not only on molecules identified as statistically significantly changed but also associated to molecular processes such as insulin resistance, cellular senescence and p53 could reinforce the Author findings, making the

paper not a description of a list of result obtained by a careful analysis but contextualized in the frame of muscle physiology, since as indicated in the paper a "stronger knowledge of their physiological functions, are critical for understanding both damage accumulation".

From the statistical point of view the use of two models, a linear regression and the standard and zero- adjusted negative binomial models (in both linear and negative binomial models) is a good strategy to overcome the limitation imposed by the number of samples, in fact 8 RNAs with the lowest p-values overlapped between the two models, confirming that these molecules can be good candidate for further studies in the context of muscle decline in aging. However the description of molecular processes in relation to muscle function is superficial and need to be implemented. Many of the functions associated to these proteins are derived from studies not directly connected to the muscle an effort should be done to contextualize all this transcript in the frame of the muscle for example the role of LPP, or SKAP2 or LGI1, FEZ2 the latter is implied in processes other that the one referred by Authors and very crucial in muscle aging. TPI1 and LDHA have been described very well in many papers associated to muscle in humans and their function should be referred to human muscle metabolism.

Another point that was not carefully considered was the p53 downregulation. Since in this paper is not a description of a new technology but focuses on results provided from the application of advanced approaches at the state of the art to human muscle of healthy aged subjects, verification of one of the main downstream effector for cell cycle, apoptosis , DNA repair all involved in senescence will provide more insight into the physiological decline of muscle mass and force typical of aging. Other points to be carefully revised are the role of PLAG1,Nudix Hydrolase 6, involved in DNA repair better that fibroblast growth or the role of ELAV 1 and MOV10.

Concerning the relationship between transcript and proteome, also in this case since the Authors have the possibility to compare a large data set from their previous deep proteomic investigation on the same subjects, I will expect a more careful and muscle oriented discussion on this comparison.

Response to the comments from Reviewer #1:

The authors have completed an extensive and very interesting analysis. The following comments are fully intended to enhance the clarity of the article and ensure that the data analysis is as comparable as possible with other studies in the literature.

Introduction

Clarification of technology utilized and its strengths and weaknesses

In the introduction RNA sequencing is put forward as an advantage of the present study over previously published microarray studies. This motivation should be tempered by empirical observations that unfortunately RNA-sequencing is neither as robust nor comprehensive as first thought while the sequencing consortium concluded that arrays and sequencing are complementary.

[AU] We thank the reviewer for pointing this out and share his/her view that RNA-seq is not as robust, comprehensive, or quantitative as it was previously thought. Still, there are many advantages of RNA-seq, such as the characterization of alternatively spliced transcripts, discovery of novel RNAs, and identification of non-coding RNAs. Moreover, the reviewer's concern about the robustness of the RNA-seq has been addressed in this study by sequencing at great depth (>80 million reads per sample) compared to <40 million reads reached in most studies. We have added a sentence to the manuscript stating that RNA-seq analysis has some advantages over microarrays and acknowledging that the two techniques are complementary.

The lack of comprehensive transcriptome coverage is a reflection of the inherent limitations of sequencing, whereby the most abundant RNA species are sequenced over and over again, while the vast majority are not quantified as robustly or in a linear manner (Lindholm et al., 2014; Sood et al., 2016; Timmons et al., 2018).

[AU] We appreciate this comment. We believe that the depth of sequencing alleviates the referee's concern, as it has allowed good coverage for even the bottom 1,000 RNAs. A coverage plot from one of the samples is given below (**Figure 1 for Reviewers**). These plots show that for the top 1,000 RNAs, the mean coverage is over 5000X (left graph), for the middle 1,000 RNAs it is about 40X (center graph), and for the bottom 1,000 RNAs it is more than 15X (right graph). It is true that the most abundant RNAs are sequenced over and over again, but even for the bottom 1,000 RNAs, the coverage was enough to identify low-expressed transcripts with confidence.

Figure 1 for Reviewers

Further, the preparation of the sequencing library results in up to 50% of the transcriptome being missed - while the majority of this transcriptome can be detected using the latest 'tiling' type arrays where the 'DNA' library prepared from the RNA covers a broader range of RNA species, not just RNA with long 3' poly-A tails (Lindholm et al., 2014; Sood et al., 2016; Timmons et al., 2018).

[AU] We thank the reviewer for pointing this out. Out of the 57,205 transcripts in the Ensembl hg19 v82 database, we identified on average $24,453.47 \pm 4610.29$ transcripts with ≥ 10 reads. The maximum number

of different RNAs identified in any sample was 36,119. However, as not all genes are transcribed in a particular tissue type, it is not possible to calculate what percentage of the entire transcriptome is captured by the sequencing analysis.

In addition to these laboratory factors, there are also additional assumptions and challenges assembling the raw data that limits data interpretation. It should be noted that high density arrays should always be re-annotated to the latest genome build and thus the retained short probes remain current and accurate.

[AU] We agree that high-density array should always be re-annotated. In this study we have only used RNA-seq analysis; as the full project is a multi-omics study spanning several years, we decided to use hg19 and Ensembl v82 (September 2015) for the whole project.

It is for these reasons that the authors would be more correct to refer to limitations of both methods, and older array technologies while to date - including the present study - a substantially greater proportion of the transcriptome is still captured in practice, with properly configured high-density 'tiling' type arrays (Timmons et al., 2018).

[AU] These points are well taken. As mentioned above, we identified on average $24,453.47 \pm 4,610.29$ transcripts at a frequency of ≥ 10 reads using RNA-seq. It is not possible to say if this number represents 50% or a higher percentage of the transcriptome, as not all genes are transcribed in a particular tissue type. RNA-seq is certainly not a perfect technology, but the fact that many of the transcripts found to be strongly correlated with aging had already been identified by arrays and other technologies in previous studies increase our confidence on the robustness of our results. We also found that other signals that were never previously identified, but for which there is a strong biological rationale for being affected by aging. It is exciting to find that completely different subject populations analyzed with completely different technologies produce such a strong overlap of results. This discovery supports the value and impact of our findings and underscores the importance of expanding upon the functions of these genes. We expand on this point in the Discussion section.

This will be true also for the splicing analysis where the authors present 4 robust splicing events in the present analysis - somewhat less than reported elsewhere. It should also be made clear in the abstract that many comparisons were based on 12 younger subjects and 5 older subjects which is insufficient to generate robust statistical analysis

[AU] The reviewer brings up another good point. We acknowledge that we have few old individuals (>80 years) in this cohort, as having healthy older subjects according to our stringent criteria was challenging. For example, the reviewer likely recognizes the difficulty of recruiting healthy individuals above the age of 80 without a history of knee surgery. Had we been able to recruit a larger group, we may have had more age-associated splicing events. In the revised text, we make explicit note of these limitations.

Study population

A second strength of the present study over and above existing studies is that access to very healthy individuals enrolled according to strict clinical and functional criteria was possible. While this is undoubtedly correct for many of the smaller studies carrying out molecular profiles in human muscle, the present study did not present metabolic or cardiovascular capacity measures to clarify the physiological status of the subjects while several other studies of human muscle age have done so.

[AU] The Reviewer makes an excellent request. In the original description, we stated that this study enrolled participants who were healthy based on very strict criteria that included a detailed medical history, a physical exam performed by a research nurse practitioner, a series of blood tests, an EKG, and an in-depth examination of physical and cognitive performance. This information was collected during a 5-hour medical examination in our research clinic, using a set of standard criteria described in the paper. To address the

reviewer's request, we have now updated Supplementary Figure 1 with the following information: education, BMI, waist circumference, systolic blood pressure, resting metabolic rate, VO₂ max, baseline glucose, 400 m walking time (note that the ability to walk 400 m without symptoms was one of the inclusion criteria), and knee strength (see updated table below). In a foot note, we also indicated that all study participants were free of major chronic diseases, including cardiovascular diseases, cancer and diabetes.

Characteristics of GESTALT participants in the muscle transcriptomics study sample.

Characteristic	All (n = 53)	Age ≤ median* (n = 28)	Age > median (n = 25)	p†
Male, n (%)	33 (62.3)	17 (60.7)	16 (64.0)	1
Black, n (%)	12 (22.6)	10 (35.7)	2 (8.0)	.022
Education (highest grade completed), n (%)				.457
>12	30 (56.6)	14 (50.0)	16 (64.0)	
>16	13 (24.5)	7 (25.0)	6 (24.0)	
BMI, mean (sd)	25.8 (2.8)	26.1 (2.6)	25.5 (3.0)	.414
Waist circumference‡, mean (sd)	88.3 (9.6)	85.6 (7.8)	91.3 (10.7)	.039
Systolic blood pressure, mean (sd)	119 (13)	116 (10)	122 (14)	.066
RMR‡, mean (sd)	1372 (233)	1432 (187)	1304 (265)	.066
VO ₂ max‡, mean (sd)	29.0 (6.2)	32.0 (5.9)	25.6 (4.6)	<.001
Fasting glucose, mean (sd)	89.2 (8.6)	86.8 (6.0)	91.8 (10.3)	.037
400m walk time (s), mean (sd)	250 (30)	241 (21)	261 (35)	.015
Knee strength∅ (Nm), mean (sd)	191 (63)	213 (58)	167 (61)	0.007

* The median age in the study sample is 52, 4 participants are 52 years old. The age range in the study sample is 22 – 83.

† Fisher's exact χ^2 test or t-test.

‡ $n_{\text{waist circumference}} = 51$, $n_{\text{RMR}} = 47$, $n_{\text{VO}_2\text{max}} = 42$.

∅ Left quadriceps 30 deg/s concentric peak torque. Right quadriceps 30 deg/s concentric peak torque used for one participant (31 yr). One participant (28 yr) with no quadriceps data available was removed.

It may be more prudent to reflect, in the introduction, on the full range of major studies exploring muscle age in humans (combined with transcriptome measures) to better highlight the advantages of the present study. It is note worth that the author have not cited two of the largest and most comprehensive studies, both of which relied on well defined subjects i.e. Phillips et al PLOS Genetics 2013 and Sood et al Genome Biology 2015. The former utilized a BBSRC funded cohort of healthy young. Middle-aged and older subjects that were extensively characterized.

[AU] We thank the reviewer for these suggestions and apologize for having neglected to cite these earlier studies. We now mention them in the Introduction and Discussion sections in the context of our findings.

The authors report that "knee strength (N m) increased over the first three age groups [Group 1 (20-34 yr, n=12), Group 2 (35-49 yr, n=11), and Group 3 (50-64yr, n=12)] and decreased beyond Group 3 [Group 4 (65-79 yr, n=13) and Group 5 (80+ yr, n=5)]. This is stated as being the expected trend in a healthy population but to me this seems like an unusual pattern and one that is not consistent with the accepted muscle physiology responses to age.

For example, the analysis by Lindle et al <https://www.ncbi.nlm.nih.gov/pubmed/9375323> or Kemmler et al <https://www.ncbi.nlm.nih.gov/pubmed/30443219> indicate that muscle strength is typically stable from 20y to 50yr and then declines. This suggests that the small subgroups of individuals in the present study are not

necessarily representation of larger healthy aging populations and it is strongly recommended that knee extensor is utilised as a covariate in the present linear analysis (particularly given the fact that muscle function itself is robust altered by physical training, to some extent independent of age).

[AU] The reviewer makes another important comment. In the paper from Lindle et al., muscle strength appears to be stable up to the age of 50. However, this is not the age trajectory generally described in the literature, where muscle strength is typically reported to increase up to the age of around 35-40 and then declines, with accelerated decline after the age of 70 in unselected populations. Of note, the decline is quasi-linear after the age of 40 in healthy populations. See for example the work of Dodds and colleagues, reporting that there is increase in strength after the age of 20 (**Figure 2 for Reviewers**).

Figure 2 for Reviewers

Doods RM et al. Age Ageing. 2016 Mar; 45(2): 209–216.

Doods RM et al. PLoS One. 2014 Dec 4;9(12):e113637

Of course, discrepancies can arise as different muscles behave differently, different measurement methods provide different results, and there is great individual variability in the age for which strength begins to decline. One of the possible confounding factors in the Lindle et al. study is that it assumes that the strength of every participant starts to decline at the same time, which is highly unlikely. To address this issue, we plotted knee extension strength assessed in GESTALT participants as a continuous variable according to age. We then selected participants of the Baltimore Longitudinal Study of Aging (see reference below), a study of normative aging conducted in Baltimore since 1958, matched for age, sex, and race to the GESTALT cohort. It is important to note that we used data from the first visit because the inclusion criteria of GESTALT and BLSA were exactly the same. The results are summarized below:

Moore AZ et al. Difference in muscle quality over the adult life span and biological correlates in the Baltimore Longitudinal Study of Aging. J Am Geriatr Soc. 2014 Feb;62(2):230-6. doi: 10.1111/jgs.12653. Epub 2014 Jan 17.

Figure 3 for Reviewers. Given the small sample sizes, there is substantial variability, but the two trajectories with age look very similar. Thus, within the limit of variability of a small sample, we believe that GESTALT is representative of the healthy population. We think that it would be excessive to report these data in the manuscript, but we have added language in the text to describe this analysis.

Concerning the questions about strength adjustment, we respectfully disagree with the reviewer. Adjusting for muscle strength in the regression equation would likely remove the transcriptomic components that affect muscle strength and may be responsible for the decline in muscle strength with aging. These effects are by far the most interesting, because they affect functional outcomes.

Results

The authors used the Illumina HiSeq 2500 sequencing system at a depth of >80 million single-end reads and detected, above a threshold of 10 reads per 80million, to identify 57,205 RNAs. It would be helpful to clarify how many unique genes this represented.

[AU] As mentioned above, out of the 57,205 RNAs in the Ensembl hg19 v82 database, we identified on average $24,453.47 \pm 4,610.29$ transcripts at a coverage depth of ≥ 10 reads. The maximum number of RNAs identified in any sample was 36,119. If we use a cut-off of 1 read, then on average we identified $41,068.25 \pm 7,321.32$ (≥ 1 read) with a maximum of 53,773. This information is now included in the Methods section of the paper.

Their combined model identified 506 RNAs (again gene numbers would be useful). The authors present only the top 20 genes and it would be instructive to the readers to understand how many of the 506 RNAs were reported by 4 other major studies (Melov et al., 2007; Phillips et al., 2013; Sood et al., 2015; Timmons et al., 2019).

[AU] We appreciate this suggestion. We annotated the microarray results with the latest version of the annotation from the respective companies and overlapped them with our list (476 gene symbols). We had to use gene symbols, as ensemble IDs were not available for most of the microarray results. As the reviewer can appreciate, there is appreciable overlap among the different studies. A venn diagram and table are provided to display this overlap (**Figure 4 for Reviewers**). Furthermore, among these 5 studies (including our study), there is a handful of transcripts that overlap. In particular, 2 RNAs were detected as important by all 5 studies (*LGI1* and *FEZ2* mRNAs), 11 RNAs were identified by 4/5 studies (*CAMTA1*, *COL4A5*, *KLF5*, *CYP1B1*, *USP54*, *CTH*, *PLAG1*, *NT5C2*, *TPI1*, *PFKFB2*, and *UNC13C* mRNAs), 44 mRNAs were identified by 3/5 studies, and 228 mRNAs were identified by 2/5 studies. Thus, considering all 5 studies together, our findings were replicated for a list of 285 genes and a list of 82 genes identified as significantly changing with age in our study were replicated from a at least one other study.

Figure 4 for Reviewers

Paper	Melov, 2007	Phillips, 2013	Sood, 2015	Timmons, 2019	Tumasian III
Melov, 2007	586	70	15	59	25
Phillips, 2013	70	566	38	101	42
Sood, 2015	15	38	596	32	21
Timmons, 2019	59	101	32	666	43
Tumasian III	25	42	21	43	502

It would also be useful to define how many are related to exercise status by cross referencing with these two robust exercise transcriptome signatures (Keller et al., 2010; Phillips et al., 2013).

Consideration of pathway level and gene-id level overlap will enable a more complete consideration of this questions. It is particularly important given the trend for middle-aged subjects to have greater muscle function (in males) in the present study.

[AU] We appreciate the reviewer’s excellent suggestion. A venn diagram and table displaying the overlap with the two exercise transcriptome signatures is below (**Figure 5 for Reviewers**). The diagram shows that there is some overlap between transcripts that changed with exercise in Phillips and Keller studies and those that changed with aging in our study. In particular, there were 8 RNAs associated with aging in our study that were identified as impacted by exercise by both the Phillips and Keller study (*TSPAN2, SKAP2, CAV1, NID1, JAM2, EDNRA, ABCG1, and GLCC1*), and 56 genes that were identified as impacted by exercise by at least one of these two studies. These findings suggest that some of the changes in expressed RNAs in aging muscle may be due to the well-known reduction of physical activity that occurs with aging. In our study, we attempted to minimize the impact of differences in the level of physical activity adjusting for self-

reported level of physical activity, but we cannot exclude that residual confounding factors remain. On the other hand, there is clear evidence in the literature that older persons who remain highly physically active experience much lower decline of muscle strength and mass with aging.

Figure 5 for Reviewers

Paper	Keller, 2010	Phillips, 2013	Tumasian III
Keller, 2010	856	189	30
Phillips, 2013	189	1,781	42
Tumasian III	30	42	502

qPCR validation.

The authors might wish to reflect on the 50% validation rate for the qPCR carried out on the top 20 regulated genes. They used GAPDH as the house keeping gene, which is a gene within an age modulated metabolic pathway (glycolysis). The authors own data indicate this pathway is modulated and this may be one reason for the low validation rate.

[AU] We appreciate this suggestion. The authors fully agree with the shortcomings of using *GAPDH* mRNA as a normalization control. Unfortunately, we do not have additional muscle biopsy material from this cohort to expand the qPCR analysis. We recognize this limitation in the revised manuscript.

The other consideration is that qPCR applied to only the top 20 is not really a “validation” of global transcriptomics list, as the vast majority of the 506 will not have the same statistical significance as the top 20 (i.e. it's the best case scenario). It would be more robust to compare the 506 directly with other major studies in the field - which provide supplements with the full gene lists. Meanwhile I would not use the term ‘validate’ if you are referring to validation of your 506 list.

[AU] The authors fully agree that this is a limitation of this study that was not recognized in the original text. We have revised the text to reflect the limitation of samples available for validation.

Pathway analysis

The Ingenuity Pathway analysis (and other such ontology analyses) require correction for technology (database) and tissue bias before using p-values generated by the software. IPA allows you to upload the list of 'detected' genes in your experiment to use as the background 'sampling' universe for calculating p-value enrichment values. Without this correction, the pathway analysis p-values are not correct.

[AU] We thank the reviewer for pointing this out. We have performed a new pathway analysis as suggested by the reviewer. Adipogenesis remains a significantly changed pathway even after extended testing in IPA (**Figure 6 for Reviewers**).

Figure 6 for Reviewers

Splice variant section

Please clarify why 0.1 FDR was selected - given the problems with false positive data for splicing modeling, and if the linear regression model that reports 5325 spliced RNAs related to age at $p < 0.05$ is an unadjusted p-value? If it is, please report how many were significant after correction for multiple testing. A similar issue regarding GSEA and tissue sampling bias exists when the input list is based on an unadjusted p-value (it becomes a random sample of muscle expressed genes, which will reflect muscle biology over and above a random sample of the genome).

[AU] We thank the reviewer for this remark. We used an uncorrected $p < 0.05$ in the analysis, as correction for multiple testing (Benjamini–Hochberg) drastically reduced the number of significant transcripts, probably due to the small sample size ($n=5$). In the revised manuscript, we have used a more stringent $p < 0.01$ as in the rest of the manuscript. We have updated Supplementary Figure 12 and have removed the two splice variants which did not pass this more stringent cutoff.

Discussion

A supplementary XL file with all regulated genes would greatly enhance the usefulness of the present analysis for other laboratories and should be included. Along with gene ID the precise ENST for the RNA species detected should be included.

[AU] We thank the reviewer for this valuable suggestion. We are including the table with all annotations including all gene IDs and the associated ENSTs.

For example, 7 (LGI1, SKAP2, CFAP61, NR2F2, FEZ2, PLAG1 and PFKFB2) from your top 20 in Figure 3 were reported by Timmons et al 2019 including as you mention SKAP2. Interestingly SKAP2 was first reported by Sood et al 2015 and this article should be cited.

The Sood classification model of muscle age has been re-explored and for muscle tissue age unquestionable re-validated in 2019 (See online supplement in the following new article -

<https://genomebiology.biomedcentral.com/articles/10.1186/s13059-019-1734-z>

[AU] We thank the reviewer for his/her careful comparison of our study with other datasets. We are including and discussing the references in question. We note that the manuscript text has already exceeded the maximum length specified by the journal, so we have kept our discussions succinct.

Response to the comments from Reviewer #2:

This is a very interesting research. The specific goal of this RNA-seq study reported here is to elucidate differences in the transcriptomic network of skeletal muscle as a function of age. I have some concerns on data analysis, especially on splice variant analysis.

1. Analysis tools and dataset

- *Can you deposit this dataset in a public repository such as GEO or SRA?*
- *Can you provide more information and analysis metrics on your RNA-seq libraries as a supplementary table, such as RIN, library size, mapping rate and etc? What are your criteria for inclusion/exclusion of a sequenced sample on your analysis?*
- *For open source tools such as STAR, subread, edgeR, DEXSeq, R package MASS, and pscl, what are the exactly versions you are using? Please always specify the version numbers.*
- *What are your reference genome and transcriptome in quantification?*
- *For isoform (splice variant) quantification, what tool (and its version) is used?*

[AU] We thank the reviewer for these requests. We have deposited the datasets in GEO (GSE129643). Attached are the sequencing summary and the RIN numbers. We have used STAR v2.4.0j, featurecounts v1.4.6-p5, R 3.5, EdgeR (multiple versions – 3.22.5), and DEXseq v1.26. The reference genome was hg19 and Ensembl v82 annotation was used. The differential exon usage analysis was performed using DEXSeq. We have included the versions of all software used in the revised text.

2. Genes of interest

In this paper, the authors focused on only those genes consistently go up or down with age. Surely, other patterns of expression can be very interesting as well, for example, first go-up and then go-down; or first go-down and then go-up. Of course, we cannot use a simple linear model to identify genes with such patterns. Would it be possible to detect such genes?

[AU] We thank the reviewer for this suggestion, as changes in RNA expression with age commonly follow non-uniform patterns. However, addressing this question is quite difficult. A much larger sample is needed to justify that an observed non-linear trend in expression is in fact meaningful. We believe that using linear and negative binomial models is more cautious for a study of this nature.

3. Regression model

- *For linear regression, $\log_2(\text{CPM})$ is used if I understand correctly. If so, can you justify the benefits of $\log_2(\text{CPM})$ over CPM?*
- *It's a common practice to add an offset in \log_2 transformation. Surely, the right offset is debatable. The default offset in edgeR is very small. Assume a sample has a total of 100M counted reads, and one gene (or transcript) has a single read mapped to it. Therefore, its corresponding $\log_2(\text{CPM})$ is $-6.6 (= \log_2(0.01))$. In practice, the # of counted total reads are typically between 30M and 60m in a sample, therefore, the minimal $\log_2(\text{CPM})$ is usually >-5.6 for an expressed gene. For $\log_2(\text{CPM})$, any negative number smaller than -5.6 is not biologically meaningful. Without a reasonable offset in \log_2 transformation, a very large negative number can be generated, and this in turn will cause trouble for regression. For instance, the 2nd transcript of ESRRG in one sample (in Supp Figure 11) has a $\log_2(\text{CPM}) < -10$, far away from all other samples. I strongly suggest to add "1" as the offset in \log_2 transformation. The default offset edgeR is not ideal for regression.*
- *Gene expression = sum (Transcript expression). If so, I have difficulty in harmonizing the gene expression of CFAP61 in Figure 2 with its corresponding transcript expressions in Supp Fig 11.*

[AU] We thank the reviewer for these suggestions. In our initial analysis, we used an offset (prior count) of 0.25 (the default value in edgeR), which is relatively small as the reviewer pointed out. However, for RNAs covered with ≥ 10 reads, choosing between offsets of 0.25 and 1 make negligible differences in the analysis.

For example, we used an offset (prior count) of 1 as suggested by the reviewer and compared the top RNA lists based on the two offset values. For the UP RNA list, there were 20/20 (100%) RNAs in common between the default offset and the offset of 1; for the DOWN RNA list, there were 16/20 (80%) RNAs in common between the default offset and the offset of 1. As the changes are minor, the authors would prefer to keep the default offset in the manuscript.

We also want to explain the situation for one $\log_2(\text{CPM})$ value of -10.09 (for ESRRG) in Supplementary Figure 11. We found out that the mean reads assigned to all transcripts across the samples were 273 million and EdgeR used the formula $\log_2(0.25/273) = -10.09$ (where 0.25 is the prior count) for RNAs with read counts of 0.

4. Splice Variant and Exon Usage Analyses

Lior Pachter had an interesting post (<https://liorpachter.wordpress.com/tag/differential-expression/>) on different kinds of differential expression. Isoform switches can occur without gene-level differential expression. Differential transcript usage is a question sort of independent of differential transcript expression. Changes on both usage of isoforms (resulting from alternative splicing) and expression of isoforms with age are biologically interesting. May I suggest a more comprehensive analysis of this section?

- Perform isoform-level expression analyses as you have already done at the gene level, and then compare the results. Typically, gene expression is dominated by a single or a couple of isoforms.*
- Perform transcript usage analysis, and identify genes with potential isoform switches. What are those genes and what they do? It's most likely you will get more insights from this RNA-seq dataset.*
- Too my knowledge, DEXSeq can detect exon inclusion and exclusion only, but not other kinds of splicing events, such as alternative 5-end splicing site, alternative 3'-end slicing event, or alternative polyadenylation. Some tools are particularly designed for alternative splicing, including rMATs, MAJIQ, leafcutter and etc. My gut feeling is that there are a lot of interesting changes in splicing during aging. Such splicing events can be potential predictive biomarkers, and become drug targets of 'longevity'. If so, your research will impact a much broader community who are interested in ageing.*

[AU] We appreciate the reviewer's comment and we share the notion that aging is associated with changes in the splicing repertoire. There is initial evidence in the literature that splicing regulates energy metabolism and is associated with senescence. In this paper, we decided to carry out DEXSeq analysis because we wanted to have a clear picture of skipped exons. We carried out the rMAT approach too, but ultimately decided not to present it in this paper for a number of reasons: (1) our manuscript is already very long, (2) splicing is not its main focus, and (3) we do not have muscle RNA left for validation in this study, so we cannot verify the rMATs results.

Minor typo in Supplementary text – "Supplementary Fig. 5. Proteomic analysis" should be "Supplementary Fig. 7. Proteomic analysis".

[AU] Thank you, we have fixed this typo.

Response to the comments from Reviewer #3:

The paper named: "Skeletal Muscle Transcriptome in Healthy Aging" by Tumasian III, R.A et al. provide a careful analysis of transcript levels. The study identify 57,205 protein-coding and non-coding RNAs, among them 1,134 RNAs were significantly associated with age. The cellular processes involved are: cellular senescence, insulin signaling, and myogenesis as expected. Furthermore, the careful analysis of splice variants indicate that were enriched for proteins involved in oxidative phosphorylation and adipogenesis. The study is original since a previous study published many years ago on muscle proteome analysis of healthy subjects is nowadays overcome by more efficient technical approaches even though some suggestion for interpretation of present data in the frame of muscle physiology can be of some aid (particularly for alternative splicing).

Although it is true that no previous study collected detailed information in very healthy individuals enrolled according to strict clinical and functional criteria and in a quite large number of samples with 5 individuals over 80. Nevertheless, the number of sample still represent a limitation in "omic" studies and verification adopting alternative technologies, not only on molecules identified as statistically significantly changed but also associated to molecular processes such as insulin resistance, cellular senescence and p53 could reinforce the Author findings, making the paper not a description of a list of result obtained by a careful analysis but contextualized in the frame of muscle physiology, since as indicated in the paper a "stronger knowledge of their physiological functions, are critical for understanding both damage accumulation".

[AU] We agree with the suggestions of the reviewer that interpretation of the results will eventually require much more than the simple identification of age-related genes. However, these analyses would bring this paper outside its primary scope and the size limitations of the journal. Moreover, as this study population that has been selected to be extremely healthy and most metabolic and cardiovascular parameters were selected to be within normal limits, it may not be well suited for such analyses. We have discussed these points in the revised text.

From the statistical point of view the use of two models, a linear regression and the standard and zero-adjusted negative binomial models (in both linear and negative binomial models) is a good strategy to overcome the limitation imposed by the number of samples, in fact 8 RNAs with the lowest p-values overlapped between the two models, confirming that these molecules can be good candidate for further studies in the context of muscle decline in aging. However the description of molecular processes in relation to muscle function is superficial and need to be implemented. Many of the functions associated to these proteins are derived from studies not directly connected to the muscle an effort should be done to contextualize all this transcript in the frame of the muscle for example the role of LPP, or SKAP2 or LGI1, FEZ2 the latter is implied in processes other that the one referred by Authors and very crucial in muscle aging. TPI1 and LDHA have been described very well in many papers associated to muscle in humans and their function should be referred to human muscle metabolism.

[AU] We agree that the description of the top age-related transcripts was not comprehensive and not focused on muscle physiology and aging. In the revised manuscript, we have expanded the Discussion text to address these points and provide interpretation of our findings in light of published work on SKAP2, LGI1, FEZ2, TPI1 and LDHA.

Another point that was not carefully considered was the p53 downregulation. Since in this paper is not a description of a new technology but focuses on results provided from the application of advanced approaches at the state of the art to human muscle of healthy aged subjects, verification of one of the main downstream effector for cell cycle, apoptosis , DNA repair all involved in senescence will provide more insight into the physiological decline of muscle mass and force typical of aging. Other points to be carefully

revised are the role of *PLAG1*, *Nudix Hydrolase 6*, involved in DNA repair better than fibroblast growth or the role of *ELAVL1* and *MOV10*.

[AU] As the text and reference limits have been surpassed, we point the reader to further explore the roles of these RNAs.

Concerning the relationship between transcript and proteome, also in this case since the Authors have the possibility to compare a large data set from their previous deep proteomic investigation on the same subjects, I will expect a more careful and muscle oriented discussion on this comparison.

[AU] We thank the reviewer for this comment. We reanalyzed the proteomic and transcriptomic datasets, comparing all age-associated protein coding transcripts with their corresponding protein-level data. The results are summarized in Figure 7 for Reviewers and the analysis method is discussed in the methods. Correlations between proteins and transcripts ranged between -0.377 and 0.754. Most proteins important for muscle function were found to be correlated with their transcript levels (supplementary figure 18c).

Figure 7 for Reviewers

Significant ($p < 0.05$) downstream protein abundance and $\log_2(\text{CPM})$ correlations. **a** Distribution of 1227 protein-RNA correlation coefficient analysis is shown. **b** Correlation of 122 age-associated protein-RNA matches. **c**. Heatmap of 122 mRNA associated with age and their correlations with proteins. Lowest correlation mRNA (H2AZ) is shown in the inset.

Reviewers' Comments:

Reviewer #1:

Remarks to the Author:

Dear Authors I thank you for considering my comments and suggestions. I am content that you have taken the meta-analysis of your results to the next level and placed them in context with the literature. I focus below only on the RNA methodology as there are still some misunderstandings and if you could clarify them in your manuscript this would provide a service to the field. While I appreciate the method they used is the method they used - its important for the field to have a proper articulation of the limitations of short-read global RNA-seq applied to tissue as these tissue cohorts are extremely costly and precious. The format of the response below is

1st Paragraph - original review

2nd Paragraph - Authors first reply

3rd Paragraph - reviewer clarification based on text and graphs provided by author

1. In the introduction RNA sequencing is put forward as an advantage of the present study over previously published microarray studies. This motivation should be tempered by empirical observations that unfortunately RNA-sequencing is neither as robust nor comprehensive as first thought while the sequencing consortium concluded that arrays and sequencing are complementary

2. [AU] We thank the reviewer for pointing this out and share his/her view that RNA-seq is not as robust, comprehensive, or quantitative as it was previously thought. Still, there are many advantages of RNA-seq, such as the characterization of alternatively spliced transcripts, discovery of novel RNAs, and identification of non-coding RNAs. Moreover, the reviewer's concern about the robustness of the RNA-seq has been addressed in this study by sequencing at great depth (>80 million reads per sample) compared to <40 million reads reached in most studies. We have added a sentence to the manuscript stating that RNA-seq analysis has some advantages over microarrays and acknowledging that the two techniques are complementary.

3. Rev1]. I am afraid the authors are not correct on this point and this misunderstanding impacts on some of their responses below. High throughput short read global RNA-seq applied to tissues - such as muscle where 1 or 2 thousand genes are very highly expressed - is not ideally placed to study alternative splicing or identify non-coding RNAs. The limited ability of the present study to identify splicing is a reflection of the statistical challenges modeling splicing (determining which reads are allocated to which variants, when some reads reflect exons in common). The comment on non-coding RNA is not true - if the tiling array can detect 10,000 non-coding RNAs in muscle tissue while no RNA-seq study has ever been able to report more than 2-3,000 non-coding RNAs in muscle tissue. Clearly there is a problem. This problem stems from the basics of PCR and the competitive nature of quantification for sequencing. This was not an issue for DNA as abundance was never the primary goal. If your RNA to cDNA library preparation is such that non-coding RNAs are deprioritized (due to use of oligoDT), then those RNAs don't appear in the library (i.e. 'library diversity') as they should and thus can never be sequenced and quantified.

The lack of comprehensive transcriptome coverage is a reflection of the inherent limitations of sequencing, whereby the most abundant RNA species are sequenced over and over again, while the vast majority are not quantified as robustly or in a linear manner (Lindholm et al., 2014; Sood et al., 2016; Timmons et al., 2018).

[AU] We appreciate this comment. We believe that the depth of sequencing alleviates the referee's concern, as it has allowed good coverage for even the bottom 1,000 RNAs. A coverage plot from one of the samples is given below (Figure 1 for Reviewers). These plots show that for the top 1,000 RNAs, the mean coverage is over 5000X (left graph), for the middle 1,000 RNAs it is about 40X (center graph), and for the bottom 1,000 RNAs it is more than 15X (right graph). It is true that the most abundant RNAs are sequenced over and over again, but even for the bottom 1,000

RNAs, the coverage was enough to identify low-expressed transcripts with confidence. Rev1]. I am afraid the authors have misunderstood my point. You are plotting what YOU could detect - not what is expressed in muscle and could not detect and so have focused on a different point. I think you would find it useful to examine the articles mentioned - especially the supplements where studies using 100M read-alignments are compared with the HTA 2.0 profiles (In muscle and brain). There is a good appreciation that tiling type arrays better are detecting lower abundance transcripts as each transcript has a dedicated detection set of probes and is not competing with all the cDNAs in the library. So what I was referring to was that if you take 1,000 lower expressed transcripts identified in a muscle HTA 2.0 study and plots those in your RNAseq data you will probably find you can't even detect them - but if you do, that the detection is not linear with read depth (this is shown in supplement in the Sood iGEMS article). From a biology perspective, you will appreciate that many receptors and signaling genes have relatively modest expression yet can have important physiological roles.

Further, the preparation of the sequencing library results in up to 50% of the transcriptome being missed - while the majority of this transcriptome can be detected using the latest 'tiling' type arrays where the 'DNA' library prepared from the RNA covers a broader range of RNA species, not just RNA with long 3' poly-A tails (Lindholm et al. , 2014; Sood et al., 2016; Timmons et al., 2018).

[AU] We thank the reviewer for pointing this out. Out of the 57,205 transcripts in the Ensembl hg19 v82 database, we identified on average $24,453.47 \pm 4610.29$ transcripts with ≥ 10 reads. The maximum number of different RNAs identified in any sample was 36,119. However, as not all genes are transcribed in a particular tissue type, it is not possible to calculate what percentage of the entire transcriptome is captured by the sequencing analysis.

Rev1]. I am afraid the authors have partly misunderstood my point. You say you have 24,543 unique transcripts? Or do you mean 'genes'? What % of YOUR total reads are mitochondrial? The second point to make is that if over the samples studied you have $24,453.47 \pm 4610.29$ transcripts with ≥ 10 reads, then this is a large sample to sample variation. Is that a Standard deviation? What is the mean and range per sample and how do you deal with missing values per transcript/per sample? It would be good if you include all of this information in the results.

The authors used the Illumina HiSeq 2500 sequencing system at a depth of >80 million single-end reads and detected, above a threshold of 10 reads per 80million, to identify 57,205 RNAs. It would be helpful to clarify how many unique genes this represented.

[AU] As mentioned above, out of the 57,205 RNAs in the Ensembl hg19 v82 database, we identified on average $24,453.47 \pm 4,610.29$ transcripts at a coverage depth of ≥ 10 reads. The maximum number of RNAs identified in any sample was 36,119. If we use a cut-off of 1 read, then on average we identified $41,068.25 \pm 7,321.32$ (≥ 1 read) with a maximum of 53,773. This information is now included in the Methods section of the paper.

Rev1]. Thanks - this supplements the information above. Please note, that at this level an HTA 2.0 chip, properly processed (realigned to latest genome, scanned for functional probes) detects about 80,000 to 110,000 unique ENST (transcript) identifiers per muscle sample. How many of your $41,068.25 \pm 7,321.32$ (≥ 1 read) were classified as lncRNAs and what is the number of 'genes' identifiers rather than RNA identifiers?

Pathway analysis

The Ingenuity Pathway analysis (and other such ontology analyses) require correction for technology (database) and tissue bias before using p-values generated by the software. IPA allows you to upload the list of 'detected' genes in your experiment to use as the background 'sampling' universe for calculating p-value enrichment values. Without this correction, the pathway analysis p-values are not correct.

[AU] We thank the reviewer for pointing this out. We have performed a new pathway analysis as suggested by the reviewer. Adipogenesis remains a significantly changed pathway even after extended testing in IPA (Figure 6 for Reviewers).

Rev1]. Thanks - please note there is no significant Z-score and the p-value is weak. The threshold shown is not considered to be that useful for these pathway or ontology type analysis due to the inflated starting p-values that occur due to the database and sampling bias and the lack of independence of the variables (transcripts). For example, if you compared all detected gene symbols from your muscle study with the genome - you would see enormous p-values. So I would tread carefully with this particular result..

Reviewer #2:

Remarks to the Author:

First, thank the authors for clarification and having addressed most of my previous questions.

In the paper, "RNA" refers to "gene", and "splice variants" means "transcripts" or "isoforms". It would be much clearer to use "gene/transcript" rather than "RNA/splicing variant", at least to most audience.

Quantification of "splice variant": The authors use "featureCounts" in subread package to count reads. To my knowledge, featureCounts is only a counting tool, but cannot split reads among different "splice variants". I am wondering how the authors get the expression level for INDIVIDUAL "splice variants" using featureCounts. Usually, researchers use RSEM, Salom and Kallisto for isoform ("splice variants") quantification.

The section "Splice variants and exon usage with age" (line 225-290):

It is sort of confusing simply because the authors talk about four different kinds of changes in: (1) RNA ("gene") EXPRESSION; (2) splice variant (isoform) EXPRESSION; (3) exon level EXPRESSION; and (4) the RELATIVE usage of splice variant (isoform). It makes more sense to perform the analysis separately as below (<https://liorpachter.wordpress.com/tag/differential-expression/>)

- Differential transcript expression (DTE)
- Differential transcript usage (DTU)
- Differential exon expression

I believe more complementary insights can be gained from different differential analysis. For example, it's likely to identify "splice variants" (of the same gene) that change with age in opposite directions, but no clear change at the gene level.

Reviewer #3:

Remarks to the Author:

I thank the Author for its careful review and for answers provided point by point.

However there is a point that I missed, the novelty introduced by this study, I think it is difficult to extract what is really novel and introduced for the first time, but also what is strictly related with the decrease of muscle mass and strenght in healthy aged subjects. I will suggest to include a paragraph hilighting the novel results obtained for the first time adopting this analytical approach.

Reviewer #1 (Remarks to the Author):

The authors' responses in this further revised point-by-point response are distinguished by black font and "[AU2]"

Dear Authors I thank you for considering my comments and suggestions. I am content that you have taken the meta-analysis of your results to the next level and placed them in context with the literature. I focus below only on the RNA methodology as there are still some misunderstandings and if you could clarify them in your manuscript this would provide a service to the field. While I appreciate the method they used is the method they used - it's important for the field to have a proper articulation of the limitations of short-read global RNA-seq applied to tissue as these tissue cohorts are extremely costly and precious. The format of the response below is:

1st Paragraph - original review

2nd Paragraph - Authors first reply

3rd Paragraph - reviewer clarification based on text and graphs provided by author

1. In the introduction RNA sequencing is put forward as an advantage of the present study over previously published microarray studies. This motivation should be tempered by empirical observations that unfortunately RNA-sequencing is neither as robust nor comprehensive as first thought while the sequencing consortium concluded that arrays and sequencing are complementary

2. [AU] We thank the reviewer for pointing this out and share his/her view that RNA-seq is not as robust, comprehensive, or quantitative as it was previously thought. Still, there are many advantages of RNA-seq, such as the characterization of alternatively spliced transcripts, discovery of novel RNAs, and identification of non-coding RNAs. Moreover, the reviewer's concern about the robustness of the RNA-seq has been addressed in this study by sequencing at great depth (>80 million reads per sample) compared to <40 million reads reached in most studies. We have added a sentence to the manuscript stating that RNA-seq analysis has some advantages over microarrays and acknowledging that the two techniques are complementary.

3. Rev1]. I am afraid the authors are not correct on this point and this misunderstanding impacts on some of their responses below. High throughput short read global RNA-seq applied to tissues - such as muscle where 1 or 2 thousand genes are very highly expressed - is not ideally placed to study alternative splicing or identify non-coding RNAs. The limited ability of the present study to identify splicing is a reflection of the statistical challenges modeling splicing (determining which reads are allocated to which variants, when some reads reflect exons in common). The comment on non-coding RNA is not true - if the tiling array can detect 10,000 non-coding RNAs in muscle tissue while no RNA-seq study has ever been able to report more than 2-3,000 non-coding RNAs in muscle tissue. Clearly there is a problem. This problem stems from the basics of PCR and the competitive nature of quantification for sequencing. This was not an issue for DNA as abundance was never the primary goal. If your RNA to cDNA library preparation is such that non-coding RNAs are deprioritized (due to use of oligoDT), then those RNAs don't appear in the library (i.e. 'library diversity') as they should and thus can never be sequenced and quantified.

[AU2] We thank the Reviewer for these points and we share his/her perspective. We have first-hand appreciation of the strengths of array analysis, as we have worked extensively with Affymetrix, Illumina and Agilent arrays in the past. In this manuscript we have tried to take advantage of the specific

strengths of high-depth RNA-seq. We have used Total RNA-seq library preparation kits, which do not rely on oligo-dT primer only and RT is initiated randomly across the transcript using Ribo-SPIA technology (<https://www.ncbi.nlm.nih.gov/pubmed/16123149>) using NuGEN Ovation RNA-seq system v2. Because of this approach, non-coding RNAs were amplified at similar rates as mRNAs. Only ribosomal RNA was deprioritized in order to increase the representation of other types of RNA. In our samples, out of 57,773 RNAs in the database, we identified 49,129 RNA in muscle tissue (all samples) at a read depth of 10 or more reads (85% of RNAs database). Of this population, 19,970 reads were protein-coding mRNAs (98% of the protein-coding mRNAs in the database) and 29,129 were non-coding RNAs (77.8% of the non-coding RNAs in the database) in muscle tissue. To provide some quantification of this comparison, we include below a figure showing the number of each RNA type in database and in our study. In the revised text, we expand on these points.

The lack of comprehensive transcriptome coverage is a reflection of the inherent limitations of sequencing, whereby the most abundant RNA species are sequenced over and over again, while the vast majority are not quantified as robustly or in a linear manner (Lindholm et al., 2014; Sood et al., 2016; Timmons et al., 2018).

[AU] We appreciate this comment. We believe that the depth of sequencing alleviates the referee’s concern, as it has allowed good coverage for even the bottom 1,000 RNAs. A coverage plot from one of the samples is given below (Figure 1 for Reviewers). These plots show that for the top 1,000 RNAs, the mean coverage is over 5000X (left graph), for the middle 1,000 RNAs it is about 40X (center graph), and for the bottom 1,000 RNAs it is more than 15X (right graph). It is true that the most abundant RNAs are sequenced over and over again, but even for the bottom 1,000 RNAs, the coverage was enough to identify low-expressed transcripts with confidence.

Rev1]. I am afraid the authors have misunderstood my point. You are plotting what YOU could detect - not what is expressed in muscle and could not detect and so have focused on a different point. I think you would find it useful to examine the articles mentioned - especially the supplements where studies using 100M read-alignments are compared with the HTA 2.0 profiles (In muscle and brain). There is a good appreciation that tiling type arrays better are detecting lower abundance transcripts as each transcript has a dedicated detection set of probes and is not competing with all the cDNAs in the library. So what I was referring to was that if you take 1,000 lower expressed transcripts identified in a muscle HTA 2.0 study and plots those in your RNA-seq data you will probably find you can't even detect them - but if you do, that the detection is not linear with read depth (this is shown in supplement in the Sood iGEMS article). From a biology perspective, you will appreciate that many receptors and signaling genes have relatively modest expression yet can have important physiological roles.

[AU2] We appreciate the Reviewer's thoughtful reflection on these points. To address this specific concern, we examined the overlap of genes listed in the supplementary Table S2 from Sood et al. (2016) with our data. We found that 989 RNAs reported by Sood et al. (and present in Ensembl) were not detected in our study when we used a read depth of 10 or more reads. However, our study identified an additional 27,641 RNAs expressed in muscle tissue that not previously reported by Sood et al. (2016); of note, 410 RNAs reported in Sood et al. (2016) were not present in the Ensembl v82 database. In sum, we fully agree with the reviewer's previous statement that both microarray and RNA-seq have advantages and the two approaches can complement each other. We further emphasize this point in the revised text.

Further, the preparation of the sequencing library results in up to 50% of the transcriptome being missed - while the majority of this transcriptome can be detected using the latest 'tiling' type arrays where the 'DNA' library prepared from the RNA covers a broader range of RNA species, not just RNA with long 3' poly-A tails (Lindholm et al., 2014; Sood et al., 2016; Timmons et al., 2018).

[AU] We thank the reviewer for pointing this out. Out of the 57,205 transcripts in the Ensembl hg19 v82 database, we identified on average $24,453.47 \pm 4610.29$ transcripts with ≥ 10 reads. The maximum number of different RNAs identified in any sample was 36,119. However, as not all genes are transcribed in a particular tissue type, it is not possible to calculate what percentage of the entire transcriptome is captured by the sequencing analysis.

Rev1]. I am afraid the authors have partly misunderstood my point. You say you have 24,543 unique transcripts? Or do you mean 'genes'?

[AU2] Thank you for pointing this out. By "24,543 unique transcripts", we meant 24,543 unique RNAs i.e. transcripts from 24,543 unique genes and not splice variants. We apologize for the confusion.

What % of YOUR total reads are mitochondrial?

[AU2] We appreciate this question. In our study, $26.38 \pm 13.87\%$ of reads were assigned to mitochondrial RNA.

The second point to make is that if over the samples studied you have $24,453.47 \pm 4610.29$ transcripts with ≥ 10 reads, then this is a large sample to sample variation. Is that a Standard deviation?

[AU2] These are standard deviations.

What is the mean and range per sample and how do you deal with missing values per transcript/per sample? It would be good if you include all of this information in the results.

[AU2] As mentioned above, the average number of transcripts identified is 24,543 with a range of 16,203 to 36,119. The distribution is on the right.

As advised by the reviewer, genes having zeros in all samples were removed. For missing values, a prior count of 0.250 was added.

The authors used the Illumina HiSeq 2500 sequencing system at a depth of >80 million single-end reads and detected, above a threshold of 10 reads per 80million, to identify 57,205 RNAs. It would be helpful to clarify how many unique genes this represented.

[AU] As mentioned above, out of the 57,205 RNAs in the Ensembl hg19 v82 database, we identified on average $24,453.47 \pm 4,610.29$ transcripts at a coverage depth of ≥ 10 reads. The maximum number of RNAs identified in any sample was 36,119. If we use a cut-off of 1 read, then on average we identified $41,068.25 \pm 7,321.32$ (≥ 1 read) with a maximum of 53,773. This information is now included in the Methods section of the paper.

Rev1]. Thanks - this supplements the information above. Please note, that at this level an HTA 2.0 chip, properly processed (realigned to latest genome, scanned for functional probes) detects about 80,000 to 110,000 unique ENST (transcript) identifiers per muscle sample. How many of your $41,068.25 \pm 7,321.32$ (≥ 1 read) were classified as lncRNAs and what is the number of 'genes' identifiers rather than RNA identifiers?

[AU2] Out of 49,129 RNAs identified in muscle tissue at a depth of 10 or more reads, we identified 29,129 non-coding RNAs (77.8% of non-coding RNAs present in database). Out of these, 6,626 lncRNAs were identified (93% of lncRNAs present in database).

Pathway analysis

The Ingenuity Pathway analysis (and other such ontology analyses) require correction for technology (database) and tissue bias before using p-values generated by the software. IPA allows you to upload the list of 'detected' genes in your experiment to use as the background 'sampling' universe for calculating p-value enrichment values. Without this correction, the pathway analysis p-values are not correct.

[AU] We thank the reviewer for pointing this out. We have performed a new pathway analysis as suggested by the reviewer. Adipogenesis remains a significantly changed pathway even after extended testing in IPA (Figure 6 for Reviewers).

Rev1]. Thanks - please note there is no significant Z-score and the p-value is weak. The threshold shown is not considered to be that useful for these pathway or ontology type analysis due to the inflated starting p-values that occur due to the database and sampling bias and the lack of independence of the variables (transcripts). For example, if you compared all detected gene symbols from your muscle study with the genome - you would see enormous p-values. So I would tread carefully with this particular result...

[AU2] Yes, we fully agree with the words of caution from the reviewer, and we added text to reflect this point in the revised manuscript.

Reviewer #2 (Remarks to the Author):

First, thank the authors for clarification and having addressed most of my previous questions.

In the paper, “RNA” refers to “gene”, and “splice variants” means “transcripts” or “isoforms”. It would be much clearer to use “gene/transcript” rather than “RNA/splicing variant”, at least to most audience.

[AU2] We thank the reviewer for his/her advice, as we have tried to adopt the language most appropriate for this audience.

We would like to continue using ‘**gene**’ in the most rigorous sense of the word, as defined in textbooks: a segment of DNA that can be transcribed. There are very few exceptions to this strict definition (only RNA viruses are the exception). To continue with textbook definitions, ‘**RNAs**’ are the molecules transcribed from a DNA segment (where the ‘DNA segment’ is the ‘gene’); the RNAs that are transcribed from a given DNA segment may all be identical or may have variants, and ‘**isoforms**’ are the variant RNAs that can be transcribed from a gene through altered initiation, splicing, termination, and processing.

Following the reviewer’s advice to increase clarity and reduce disagreement between the world of classical molecular biology and bioinformatics in this manuscript, we have used “RNA” to mean “gene expression”, and we have used “transcript” when we refer to isoforms or splice variants. Being aware of the nomenclature used by computational scientists, we use “differential transcript usage (DTU)” to mean “differential isoform usage” instead of “Splice variants and exon usage with age”. In the revised manuscript, we define these terms to avoid confusion.

If the editor asks that we relax the word ‘gene’ to refer to the ‘collection of RNAs expressed from a DNA segment’, we will do that. However, using the more colloquial definitions of ‘gene’ to mean ‘RNA’ will be more confusing to the readers.

Quantification of “splice variant”: The authors use “featureCounts” in subread package to count reads. To my knowledge, featureCounts is only a counting tool, but cannot split reads among different “splice variants”. I am wondering how the authors get the expression level for INDIVIDUAL “splice variants” using featureCounts. Usually, researchers use RSEM, Salom and Kallisto for isoform (“splice variants”) quantification.

[AU2] We also thank the reviewer for this advice. The authors agree with the limitation of featureCounts when used for transcript counts, as it gives biased transcript expression counts and the highly expressed transcripts dominate the counts. Indeed, Kallisto or RSEM are far superior methods; accordingly, we have reanalyzed the transcript expression portion of the of the manuscript using Kallisto. To incorporate the reviewer’s suggestions, we have modified Figure 6, and Supplementary Figures 10-13 and 15, and replaced *FEZ2* with *TET2* in Figure 6 and Supplementary Figure 11. We also changed *NUDT6* for *RXYLT1* in Supplementary Figure 13 and 15. This change allowed us to use a stricter p -value <0.01 in Supplementary Figure 12 instead of p -value <0.05 used in the earlier version of the manuscript. Despite the more rigorous analysis, the conclusions of the study were not substantially altered.

The section “Splice variants and exon usage with age” (line 225-290):

It is sort of confusing simply because the authors talk about four different kinds of changes in: (1) RNA (“gene”) EXPRESSION; (2) splice variant (isoform) EXPRESSION; (3) exon level EXPRESSION; and (4) the RELATIVE usage of splice variant (isoform). It makes more sense to perform the analysis separately as below (<https://liorpachter.wordpress.com/tag/differential-expression/>)

- Differential transcript expression (DTE)
- Differential transcript usage (DTU)
- Differential exon expression

[AU2] We thank the reviewer for this request. In this manuscript we have used age as a continuous variable and most of the DTU programs use only two groups. Hence, we have used and simplified method by using percentage of each transcript and linear regression of the percentages (Figure 6). We also expanded the section on “**Splice variants and exon usage with age**”. To be mindful of the nomenclature used by computational scientists, we are using “differential transcript usage (DTU)” to mean “differential isoform usage”. This definition is provided for the reader.

I believe more complementary insights can be gained from different differential analysis. For example, it’s likely to identify “splice variants” (of the same gene) that change with age in opposite directions, but no clear change at the gene level.

[AU2] We appreciate this excellent comment. We overlapped the RNAs that did not change significantly with age in either the linear or NB model and found two transcripts with significantly changed usage with age. We selected two examples with high beta values for the transcripts (*SELENOF* and *SLC47A1*) and included them in Supplementary Fig. 12. Accordingly, the original Supplementary Figures 12-17 are now Supplementary Figures 13-18.

Reviewer #3 (Remarks to the Author):

I thank the Author for its careful review and for answers provided point by point. However there is a point that I missed, the novelty introduced by this study, I think it is difficult to extract what is really novel and introduced for the first time, but also what is strictly related with the decrease of muscle mass and strength in healthy aged subjects. I will suggest to include a paragraph highlighting the novel results obtained for the first time adopting this analytical approach.

[AU2] We appreciate these comments. While several studies already described changes in the transcriptome associated with aging, none of the previous studies focused on a population of healthy individuals screened according to very strict clinical criteria. Since the prevalence of morbidity for chronic diseases as well as multimorbidity increases geometrically with aging, in an unselected population it is impossible to know whether the changes detected with aging are truly due to aging or to age-associated diseases. The fact that our results replicated some of the previous findings, and in part those that have been associated with sarcopenia and frailty, strongly suggest that accelerated muscle aging is at the root of sarcopenia and sarcopenia-related frailty. The use of RNA-seq analysis further allowed us to identify new age-related transcripts that point to still unexplored mechanisms.

In addition, we report for the first time splicing variants changing with aging that are linked to glucose metabolism, oxidative phosphorylation and TORC1-modulated pathways, suggesting that these variants emerge to adapt to age-related changes in energy metabolism. Although this hypothesis requires further testing, it is consistent with growing evidence from animal models that the modulation of splicing may be used to adapt energy production and utilization in the face of declining mitochondrial function due to aging or disease. In the revised manuscript, we point out to these novel findings of our report.

Reviewers' Comments:

Reviewer #1:

Remarks to the Author:

Ferrucci et al - "Skeletal Muscle Transcriptome in Healthy Aging": "The specific goal of the study reported here is to elucidate differences in the transcriptomic network of skeletal muscle as a function of age. Earlier large-scale studies identified changes in gene expression associated with skeletal muscle aging and acute exercise in well-defined populations using microarrays. Other studies have identified genomic and proteomic signatures associated with skeletal muscle aging. However, such analyses were limited by factors such as availability of probes on assay platforms, validation of results, clinical inclusion criteria, and the sensitivity and accuracy of the assays"

Study Summary

Sample size: n=53 (22-83y) small

Physiological phenotyping: Multiple - Excellent, not unique

Number of transcripts reliably detected: 24,543 <<prior art

Statistical methods: Standard, not reliable for splicing

Informatic methods: Standard, not novel

General concerns raised due to response on coverage

When the authors have replied to my concerns about RNA coverage, they have incorrectly reported literature and database values, mixed incomparable terms (e.g. total counts vs transcript vs gene). The values you used to respond to my query were wrong so I will try again to clarify the errors you are making regarding the key metrics, as they are central to the manuscript, the validity of your down-stream analysis and correct reporting of the performance of RNA detection technology.

Despite the authors reporting they have experience with various array platforms they don't appear to appreciate that Affymetrix exon or 'tiling' type arrays are near complete representations of the transcriptome, and that each 25-mer probe can be routinely updated to include/exclude in a 'transcript' build and modeled for splicing. In that sense their accuracy is only a matter of using the technology properly. As for sensitivity, its understood the dedicated probe detection approach vs competitive non-linear sequencing, provides an advantage or at least parity.

[AU2] We thank the Reviewer for these points and we share his/her perspective. We have first-hand appreciation of the strengths of array analysis, as we have worked extensively with Affymetrix, Illumina and Agilent arrays in the past. In this manuscript we have tried to take advantage of the specific. In our samples, out of 57,773 RNAs in the database, we identified 49,129 RNA in muscle tissue (all samples) at a read depth of 10 or more reads (85% of RNAs database). Of this population, 19,970 reads were protein-coding mRNAs (98% of the protein-coding mRNAs in the database) and 29,129 were non-coding RNAs (77.8% of the non-coding RNAs in the database) in muscle tissue. To provide some quantification of this comparison, we include below a figure showing the number of each RNA type in database and in our study. In the revised text, we expand on these points.

You report detecting 98% of the protein database and 93% of the noncoding database are extremely worrying – yet these values should have alerted you to the fact you must have errors in your calculations or methods as muscle does not express 98% of all protein coding genes and 93% of all noncoding genes. That could be illogical. I do not understand how you could put these values forward as a reasonable response.

There are actually 227,980 transcripts in ENSEMBL

(https://www.ensembl.org/Homo_sapiens/Info/Annotation) representing at least 20,499 are protein coding genes, 15,200 are pseudogenes (regulatory as decoy transcripts etc) and 23,992

non-coding (which is considered to be a 2-3 fold under estimation). The % detected from "the database" you present are unfortunately wrong.

First, "49,129 RNA in muscle" is not a valid representation of your data. I assume you mean 49,129 individual transcript IDs and that a transcript is counted even if – by chance – appearing in 1 clinical sample? Later state your "transcript count is 24,543 on average (with a range of 16,203 to 36,119 per sample)". This is the correct transcript count but still does not convey the number of genes (a key parameter for your pathway analysis).

So you have 24,543 ENSTs (or unique transcript IDs) reliably detected in n=52 muscle samples not 49,129. This represents 10.7% of the database transcriptome count (227,980) and not the overly optimistic values you present in your earlier reply. This value of 24,543 is only 25% of that value reported by Sood and colleagues in several muscle HTA 2.0 array studies (2016 – 2019).

Sood et al article reports >515,000 Exons detected while the available raw data and subsequent studies using the HTA 2.0 array with muscle tissue at GEO indicate they have transcript counts reliably detected in every sample of >100,000 ENSTs (i.e. multi-splice variants for each gene), which is far in excess of the present study. The custom map (CDF) has coding and noncoding 164,993 ENTs for the muscle experiments (See their Figure S8, 2019 Timmons et al Aging Cell article) which is the equivalent to your "49,129" value. Thus, if your sequencing data is inferior to the coverage with the HTA 2.0 then you do not address one of the main points of your study (See above, quote from your introduction).

[AU2] We appreciate the Reviewer's thoughtful reflection on these points. To address this specific concern, we examined the overlap of genes listed in the supplementary Table S2 from Sood et al. (2016) with our data. We found that 989 RNAs reported by Sood et al. (and present in Ensembl) were not detected in our study when we used a read depth of 10 or more reads. However, our study identified an additional 27,641 RNAs expressed in muscle tissue that not previously reported by Sood et al. (2016); of note, 410 RNAs reported in Sood et al. (2016) were not present in the Ensembl v82 database. In sum, we fully agree with the reviewer's previous statement that both microarray and RNA-seq have advantages and the two approaches can complement each other. We further emphasize this point in the revised text.

You state that Sood and the present study report similar numbers of transcripts and present a Venn diagram. You chose to use 'transcripts' for your numbers (and not the reliable, average, count) and the smaller 'gene' count for Sood et al !!– this is a very unfortunate "accident". Sood et al does NOT report 22,887 transcripts in muscle – this is "genes" detected consistently across all samples. So to compare with your 49,129 "entities" (which should actually be your transcript count of 24,543 on average (with a range of 16,203 to 36,119) the value should be 164,993 ENTs.

For example, reading the methods from the same group as Sood, e.g. Timmons et al Aging Cell 2019, a study that is 10 times larger than your present biopsy study (n=577 muscle samples, phenotyped for VO2max, BMI, Insulin sensitivity etc. See Figure 1 in their article) and with similar aims.

Even when combining common ENSTs from HTA 2.0 and U133+2 (both using custom CDF with common ENST probe-sets) they still detected 73,645 unique protein coding transcripts in muscle That is at least 29% more transcripts than the present study reports in total, while the HTA 2.0 data sets (GSE104235) report ~ 4x more transcripts (variants) than the U133+2, including more coding and more noncoding.

So this 2019 study provides more coverage of RNA, greater validation of results, and includes key clinical data missing from the present study and as such your introduction does not appear to be a fair representation of the state of art of the field.

For sure, your claim that you detect "an additional 27,641 RNAs expressed in muscle tissue that not previously reported by Sood et al. (2016)" is completely false because you are counting transcript numbers for your study and only gene ids for Sood.

[AU2] Thank you for pointing this out. By "24,543 unique transcripts", we meant 24,543 unique RNAs i.e. transcripts from 24,543 unique genes and not splice variants. We apologize for the confusion.

So this is your 'reliably' detected transcript number? But still you have not detailed what the reliable genes count was.

All of this confusion would have been avoided had you presented for review a summary XL file of the counts per transcript per sample, the final transcript count reliably detected and the final gene count (and categories) reliably detected. It will show that you have 25-50% coverage compared with the studies using HTA 2.0 arrays. It should be easy to produce a simple table with the following headers. Please provide an XL file of these data.

Summary Values

Total Gene IDs Protein-coding Gene IDs noncoding Gene IDs

Mean

SD

Total Transcript IDs Protein-coding ENST IDs noncoding ENST IDs

Mean

SD

As well as provide Raw counts per gene per sample, and then show the decisions made to set a reliable detection threshold, e.g. X% of n=52 had counts above X. This will establish how you motivate the final 'reliable' transcript and gene counts that would be required to run any statistical analysis.

Non-code comment

[AU2] Out of 49,129 RNAs identified in muscle tissue at a depth of 10 or more reads, we identified 29,129 non-coding RNAs (77.8% of non-coding RNAs present in database). Out of these, 6,626 lncRNAs were identified (93% of lncRNAs present in database).

These claims are very worrying. 6,626 is NOT 93% of the noncoding transcriptome by any estimate. I recommend the authors have an independent bioinformatician check all their analysis and workings. See here for a very conservative estimate for the noncoding RNA counts (https://www.ensembl.org/Homo_sapiens/Info/Annotation). Try the noncode consortium for more comprehensive figures.

Splicing issues

Given the confusion over the number of transcripts reliably detected, I was concerned about the claims made regarding splicing events. It is accepted that simple linear regression models, as used by the authors, is not a robust genome wide approach to identifying splice variants (See any review on splicing and genome wide methods) and while the authors focus on 5 candidates, it is very unclear how many of the ~1,100 "splice variants" they report are real.

We already know something is clearly wrong with tissue sequencing modelling, and this 'something' is well articulated in the literature e.g. data modelling and data quality issues - See

Zhang C, Zhang B, Lin L-L, Zhao S. Evaluation and comparison of computational tools for RNA-seq isoform quantification. *BMC Genomics*. 2017;18(1):583, and citations within. In short, sequencing aspires to address splicing but so far, when applied to clinical tissue cohorts, it fails to live up to the hype. See Westoby et al 2020, for an update on the further issues with more advanced RNAseq methods.

The authors therefore need to reflect on the general failure of tissue RNAseq for splicing analysis e.g. Scott et al used RNAseq in a much larger muscle study than the present work (n=271) and yet the data quality is very poor. Despite claiming to be 'transcriptome wide' they report 1 splice candidate. Compare this with the performance of the splicing analysis presented by Sood et al 2016 (*Nucleic Acids Res*. 2016 Jun 20;44(11):e109. doi: 10.1093/nar/gkw263.) where thousands of splicing events are reliably detected across multiple human tissues (4,421 events). GTEx has a similar issue – reporting very little splicing across tissues (e.g. Mele 2016).

While there is no doubt sequencing (including 3' & 5' RACE) is critical for studying transcript variation in detail, the idea that its ideal for discovering the landscape of splicing in clinical cohorts is simply not yet correct. Furthermore, the tiling arrays have been able to study individual exons and splicing junctions for nearly a decade and so it is very misleading to claim that your RNAseq approach is doing something that was 'not possible' before.

Pathway analysis issues

Rev1]. Thanks - please note there is no significant Z-score and the p-value is weak. The threshold shown is not considered to be that useful for these pathway or ontology type analysis due to the inflated starting p-values that occur due to the database and sampling bias and the lack of independence of the variables (transcripts). For example, if you compared all detected gene symbols from your muscle study with the genome - you would see enormous p-values. So I would tread carefully with this particular result...

[AU2] Yes, we fully agree with the words of caution from the reviewer, and we added text to reflect this point in the revised manuscript.

If you fully agree you would remove anything that does not have a significant Z score? Given that you have so far not presented the correct values for the reliably 'detectable' transcriptome in your study you can not possibly have the correct transcriptomic background to feed into IPA and thus your statistics will not be reliable. You may even be missing events.

The correct list to submit to IPA as background is the gene symbols that represent the "transcript count is 24,543 on average)". i.e. unless it is 1 transcript per gene in your analysis it will be a figure < 24,543.

Reviewer #2:

Remarks to the Author:

Thank you for re-analyzing the dataset and identify examples where different transcripts of a same gene change in opposite direction, with no clear change pattern at the gene level.

For isoform quantification, kallisto is a good tool.

Reviewer #1 (Remarks to the Author):

Ferrucci et al - "Skeletal Muscle Transcriptome in Healthy Aging": "The specific goal of the study reported here is to elucidate differences in the transcriptomic network of skeletal muscle as a function of age. Earlier large-scale studies identified changes in gene expression associated with skeletal muscle aging and acute exercise in well-defined populations using microarrays. Other studies have identified genomic and proteomic signatures associated with skeletal muscle aging. However, such analyses were limited by factors such as availability of probes on assay platforms, validation of results, clinical inclusion criteria, and the sensitivity and accuracy of the assays"

Study Summary

Sample size: n=53 (22-83y) small

Physiological phenotyping: Multiple - Excellent, not unique

Number of transcripts reliably detected: 24,543 <<prior art

Statistical methods: Standard, not reliable for splicing

Informatic methods: Standard, not novel

General concerns raised due to response on coverage

When the authors have replied to my concerns about RNA coverage, they have incorrectly reported literature and database values, mixed incomparable terms (e.g. total counts vs transcript vs gene). The values you used to respond to my query were wrong so I will try again to clarify the errors you are making regarding the key metrics, as they are central to the manuscript, the validity of your down-stream analysis and correct reporting of the performance of RNA detection technology.

Despite the authors reporting they have experience with various array platforms they don't appear to appreciate that Affymetrix exon or 'tiling' type arrays are near complete representations of the transcriptome, and that each 25-mer probe can be routinely updated to include/exclude in a 'transcript' build and modeled for splicing. In that sense their accuracy is only a matter of using the technology properly. As for sensitivity, its understood the dedicated probe detection approach vs competitive non-linear sequencing, provides an advantage or at least parity.

[AU3] We thank the Reviewer for these additional comments and appreciate his/her strong predilection for Affymetrix arrays. Every technology has advantages and limitations, and Affymetrix and RNA-seq platforms are both excellent for assessing transcript levels as well as for analyzing splicing. In this study, it was not our intention to compare one platform with the other, although it would be interesting to do such a side-by-side comparison in a dedicated study in the future. The reviewer's comment prompted us to survey the platforms used in gene expression studies submitted over the past 12 months to GEO. As of 6/12/2020, there was a good number of analyses performed using Affymetrix (962 studies), but the microarray platform Agilent (1,784 studies) was also very highly represented, as was the RNA-sequencing platform Illumina (7,176 studies). Therefore, there is clearly a strong choice of Illumina for gene expression studies, and this is the platform we adopted for the present study.

[AU2] We thank the Reviewer for these points and we share his/her perspective. We have first-hand appreciation of the strengths of array analysis, as we have worked extensively with Affymetrix, Illumina and Agilent arrays in the past. In this manuscript we have tried to take advantage of the specific. In our samples, out of 57,773 RNAs in the database, we identified 49,129 RNA in muscle tissue (all samples) at a read depth of 10 or more reads (85% of RNAs database). Of this population, 19,970 reads were protein-coding mRNAs (98% of the protein-coding mRNAs in the database) and 29,129 were non-coding RNAs (77.8% of the non-coding RNAs in the database) in muscle tissue. To provide some quantification of this comparison, we

include below a figure showing the number of each RNA type in database and in our study. In the revised text, we expand on these points.

You report detecting 98% of the protein database and 93% of the noncoding database are extremely worrying – yet these values should have alerted you to the fact you must have errors in your calculations or methods as muscle does not express 98% of all protein coding genes and 93% of all noncoding genes. That could be illogical. I do not understand how you could put these values forward as a reasonable response.

[AU3] We thank the Reviewer for his/her request for clarification. Regarding the coding RNAs (mRNAs), we found representation of 98% of all coding mRNAs when looking at the *aggregate of all samples* – that is, all samples added together; the % representation was of course smaller for each individual sample. It is also important to note that we did not purify muscle fibers, so the samples tested also contained small amounts of adipose tissue, nerve tissue, connective tissue, blood, and tissue-resident fibroblasts and macrophages. The entire tissue section was used to prepare each RNA library, even if some of the non-muscle tissues/cells contributed small amounts of RNA.

Regarding the 93% of the noncoding database, we realize that the reviewer misunderstood and apologize if our description was unclear. We detected 93% of the lincRNA (long *intergenic* noncoding RNA) dataset. Ensembl v82 lists 7109 lincRNAs, from which 6626 were identified in the collection of our muscle biopsies. It is important to note that lincRNAs are a small subset of the long noncoding RNA (lncRNA) content of the cell, and lncRNAs in turn are a small subset of the vast noncoding RNA collection expressed in cells. It is estimated that >85% of the human genome is transcribed, and much of that RNA is noncoding (<https://journals.plos.org/plosgenetics/article?id=10.1371/journal.pgen.1003569>).

There are actually 227,980 transcripts in ENSEMBL (https://www.ensembl.org/Homo_sapiens/Info/Annotation) representing at least 20,499 are protein coding genes, 15,200 are pseudogenes (regulatory as decoy transcripts etc) and 23,992 non-coding (which is considered to be a 2-3 fold under estimation). The % detected from “the database” you present are unfortunately wrong.

[AU3] We appreciate this remark, and realize from this and subsequent comments that the Reviewer is doing his/her calculations using ENSTs (“transcripts/isoforms”). We apologize if this was not clear in our revised text, but we made our calculations based on ENSGs (“genes/RNAs”), which reach a total number of 57,773 in Ensembl v82. Hence, our percentage calculations are correct.

First, “49,129 RNA in muscle” is not a valid representation of your data. I assume you mean 49,129 individual transcript IDs and that a transcript is counted even if – by chance – appearing in 1 clinical sample? Later state your “transcript count is 24,543 on average (with a range of 16,203 to 36,119 per sample)”. This is the correct transcript count but still does not convey the number of genes (a key parameter for your pathway analysis).

So you have 24,543 ENSTs (or unique transcript IDs) reliably detected in n=52 muscle samples not 49,129. This represents 10.7% of the database transcriptome count (227,980) and not the overly optimistic values you present in your earlier reply. This value of 24,543 is only 25% of that value report by Sood and colleagues in several muscle HTA 2.0 array studies (2016 – 2019).

[AU3] Again we are sorry for the confusion. Wherever possible, we have restricted the word ‘gene’ to its purest definition as a fragment of DNA that can be transcribed into RNA, even though some groups use the word ‘gene’ to refer to the ‘collection of RNAs transcribed from a single gene’. In the revised text, we have added definitions for ENSG and ENST to improve clarity. The Reviewer is reminded that per sample, we

have 24,453 ENSGs on average, comprising 66,632 ENSTs (transcripts/isoforms) per sample. When we combine all 53 samples, we have 165,552 ENSTs (transcripts) in total at TPM ≥ 0.1 from 49,129 ENSGs (genes/RNA) in total at counts ≥ 10 . Based on the Reviewer's advice we are removing the aggregate value of 49,129 from the manuscript and using the mean value of 24,453. We calculated that 24,453 ENSGs were expressed out of 57,773 ENSGs (the Ensembl v82 database).

Sood et al article reports >515,000 Exons detected while the available raw data and subsequent studies using the HTA 2.0 array with muscle tissue at GEO indicate they have transcript counts reliably detected in every sample of >100,000 ENSTs (i.e. multi-splice variants for each gene), which is far in excess of the present study. The custom map (CDF) has coding and noncoding 164,993 ENTs for the muscle experiments (See their Figure S8, 2019 Timmons et al Aging Cell article) which is the equivalent to your "49,129" value. Thus, if your sequencing data is inferior to the coverage with the HTA 2.0 then you do not address one of the main points of your study (See above, quote from your introduction).

[AU3] Again, we regret that the reviewer was confused by the numbers. We detected on average 66,632 ENSTs (transcripts/isoforms) per sample. Altogether, we identified a total of 165,552 ENSTs (all 53 samples combined) at TPM ≥ 0.1 .

[AU2] We appreciate the Reviewer's thoughtful reflection on these points. To address this specific concern, we examined the overlap of genes listed in the supplementary Table S2 from Sood et al. (2016) with our data. We found that 989 RNAs reported by Sood et al. (and present in Ensembl) were not detected in our study when we used a read depth of 10 or more reads. However, our study identified an additional 27,641 RNAs expressed in muscle tissue that not previously reported by Sood et al. (2016); of note, 410 RNAs reported in Sood et al. (2016) were not present in the Ensembl v82 database. In sum, we fully agree with the reviewer's previous statement that both microarray and RNA-seq have advantages and the two approaches can complement each other. We further emphasize this point in the revised text.

You state that Sood and the present study report similar numbers of transcripts and present a Venn diagram. You chose to use 'transcripts' for your numbers (and not the reliable, average, count) and the smaller 'gene' count for Sood et al !!— this is a very unfortunate "accident". Sood et al does NOT report 22,887 transcripts in muscle — this is "genes" detected consistently across all samples. So to compare with your 49,129 "entities" (which should actually be your transcript count of 24,543 on average (with a range of 16,203 to 36,119) the value should be 164,993 ENTs.

[AU3] Again, we apologize for the Reviewer's confusion. Here, we are comparing our 24,453 genes [ENSG] vs 22,887 genes reported in the article by Sood et al. To avoid confusion across biological disciplines, we use the word "gene" to mean DNA, its most basic meaning, and we use "RNA" to refer to the molecule transcribed from the gene.

For example, reading the methods from the same group as Sood, e.g. Timmons et al Aging Cell 2019, a study that is 10 times larger than your present biopsy study (n=577 muscle samples, phenotyped for VO2max, BMI, Insulin sensitivity etc. See Figure 1 in their article) and with similar aims. Even when combining common ENSTs from HTA 2.0 and U133+2 (both using custom CDF with common ENST probe-sets) they still detected 73,645 unique protein coding transcripts in muscle That is at least 29% more transcripts than the present study reports in total, while the HTA 2.0 data sets (GSE104235) report ~ 4x more transcripts (variants) than the U133+2, including more coding and more noncoding. So this 2019 study provides more coverage of RNA, greater validation of results, and includes key clinical data missing from the present study and as such your introduction does not appear to be a fair representation of the state of art of the field.

[AU3] The reviewer is correct in saying that Timmons et al (Aging Cell 2019) analyzed a large number of samples, including a subgroup that had other useful clinical measures, such as treadmill testing and measures of insulin resistance, as well as exercise interventions studies. We looked up the studies included in the Timmons et al report, but unfortunately many of them did not specify the criteria for inclusion/exclusion of participants, and therefore we were unable to exclude the effect of chronic diseases that may interfere with the assessment of the aging effect. Omitting this information was not critical for the other studies, given that they had different scopes and goals, but it hampered efforts to compare the study by Timmons et al with ours. We designed GESTALT with the specific goal of studying aging, and we therefore enrolled a population of individuals who were healthy based on very strict clinical criteria. We performed extensive testing to ensure that, to the extent of the available medical technology, they were free of acute or chronic diseases, and we had to dismiss many potential participants who met the exclusion criteria. No other previously published study has taken this approach. Having strict health standards and expanding sample size are two valid strategies, but the two studies simply cannot be compared just based on the number of samples. In this study, we found 68,349 unique protein-coding mRNA isoforms (ENSTs), and the corresponding proteins were enriched for pathways that are important in the context of aging.

For sure, your claim that you detect “an additional 27,641 RNAs expressed in muscle tissue that not previously reported by Sood et al. (2016)” is completely false because you are counting transcript numbers for your study and only gene ids for Sood.

[AU3] Again, the reviewer has misunderstood our analysis. Some studies adopt the word ‘genes’ (DNA) to refer to expressed RNAs, as in Sood et al (2016). Also, the purpose of this study is not a rigorous comparison of platforms and the information “an additional 27,641 RNAs expressed in muscle tissue that not previously reported by Sood et al. (2016)” was provided to the reviewer only and it still holds true when comparing the aggregate of all samples. If we compare the average number of RNAs/ENSGs identified per sample (24,453) to 22,887 genes identified in Sood (2016), there are an additional 1,566 RNAs expressed per sample of muscle tissue.

[AU2] Thank you for pointing this out. By “24,543 unique transcripts”, we meant 24,543 unique RNAs i.e. transcripts from 24,543 unique genes and not splice variants. We apologize for the confusion. So this is your 'reliably' detected transcript number? But still you have not detailed what the reliable genes count was.

[AU3] Here too, it is apparent that the reviewer misunderstood our analysis. We have detected on average 66,632 ENSTs (transcripts/isoforms) per sample. Altogether, we have identified 165,552 ENSTs in total (all 53 samples combined) at TPM ≥ 0.1 .

All of this confusion would have been avoided had you presented for review a summary XL file of the counts per transcript per sample, the final transcript count reliably detected and the final gene count (and categories) reliably detected. It will show that you have 25-50% coverage compared with the studies using HTA 2.0 arrays. It should be easy to produce a simple table with the following headers. Please provide an XL file of these data.

Summary Values

Total Gene IDs Protein-coding Gene IDs noncoding Gene IDs

Mean

SD

Total Transcript IDs Protein-coding ENST IDs noncoding ENST IDs
 Mean
 SD

As well as provide Raw counts per gene per sample, and then show the decisions made to set a reliable detection threshold, e.g. X% of n=52 had counts above X. This will establish how you motivate the final 'reliable' transcript and gene counts that would be required to run any statistical analysis.

[AU3] The reviewer's suggestion is excellent. In this revision, we provide the Raw counts table for ENSGs and Raw TPM table for ENSTs for the reviewer and readers upon request. We also appreciate the suggestion for the summary tables and we are providing them as well. We are adding the summary tables as Supp Fig 1b, c.

ENST summary table:

	Protein-coding	Non-coding and others	Total
Mean	27,907.59	38,075.4	66,632.98
Standard Deviation	4,527.43	10,400.17	14,728.25
Minimum	20,954	21,768	42,928
Maximum	40,371	69,753	110,124
Total (in all samples)	68,349	97,203	165,552
Percentage	83.61	84.81	84.31

ENSG summary table:

	Protein-coding	Non-coding and others	Total
Mean	15,291.4	9,162.08	24,453.47
Standard Deviation	1,758.79	2,990.68	4,610.29
Minimum	11,517	4,686	16,203
Maximum	18,966	17,339	36,119
Total (in all samples)	19,970	29,159	49,129
Percentage	98.24	77.87	85.04

Non-code comment

[AU2] Out of 49,129 RNAs identified in muscle tissue at a depth of 10 or more reads, we identified 29,129 non-coding RNAs (77.8% of non-coding RNAs present in database). Out of these, 6,626 lncRNAs were identified (93% of lncRNAs present in database).

These claims are very worrying. 6,626 is NOT 93% of the noncoding transcriptome by any estimate. I recommend the authors have an independent bioinformatician check all their analysis and workings. See [here](https://www.ensembl.org/Homo_sapiens/Info/Annotation) for a very conservative estimate for the noncoding RNA counts (https://www.ensembl.org/Homo_sapiens/Info/Annotation). Try the noncode consortium for more comprehensive figures.

[AU3] As explained above, we meant that 6,626 lincRNA (long *intergenic* non-coding RNA) were present in the database. The summary tables suggested by the reviewer have been included in the new revision, and we clarify the total number of noncoding RNAs present in the database (114,609 non-coding and other ENSTs). Just to be sure the reviewer is clear, the vast family of noncoding RNAs annotated thus far contains the family of long noncoding RNAs (lncRNA), which itself contains the subfamily of lincRNAs. The following two tables display the breakdown of the biotypes of the 114,609 transcripts/isoforms (ENSTs) and 57,734 genes (ENSGs) from our study that are present in the Ensembl database.

ENST Biotypes:

Type	Number in database	Number identified in our samples	Percentage identified
3' overlapping ncRNA	25	24	96
Antisense	9,703	9,156	94.36
IG_C_gene	18	16	88.89
IG_C_Pseudogene	10	8	80
IG_D_gene	37	0	0
IG_J_gene	18	2	11.11
IG_J_pseudogene	3	0	0
IG_V_gene	144	144	100
IG_V_pseudogene	196	178	90.82
lincRNA	11,773	11,310	96.07
miRNA	3,110	38	1.22
miscRNA	2,049	1,167	56.95
Mt_rRNA	2	2	100
Mt_tRNA	22	0	0
Non-stop decay	58	51	87.93
Nonsense mediated decay	13,046	10,690	81.94
Polymorphic pseudogene	59	47	79.66
Processed pseudogene	10,616	9,975	93.96
Processed transcript	28,037	25,218	89.95
Protein-coding	81,745	68,349	83.61
Pseudogene	387	298	77
Retained intron	25,944	23,860	91.97
rRNA	530	2	0.38
Sense_intronic	801	791	98.75
Sense_overlapping	330	313	94.85
snoRNA	1,529	114	7.46
snRNA	1,923	169	8.79
TR_C_gene	5	5	100
TR_D_gene	3	0	0
TR_J_gene	74	0	0
TR_J_pseudogene	4	0	0
TR_V_gene	97	97	100
TR_V_pseudogene	27	26	96.3
Transcribed processed pseudogene	442	354	80.09
Transcribed unprocessed pseudogene	858	689	80.3
Translated processed pseudogene	1	1	100
Unitary pseudogene	182	168	92.31
Unprocessed pseudogene	2,546	2,290	89.95
TOTAL	114,609	97,203	84.31

ENSG Biotypes:

Type	Number in database	Number identified in our samples	Percentage identified
3' overlapping ncRNA	21	21	100
Antisense	5,273	4,899	92.91
IG_C_gene	14	14	100
IG_C_Pseudogene	9	6	66.67
IG_D_gene	37	2	5.41
IG_J_gene	18	8	44.44
IG_J_pseudogene	3	0	0
IG_V_gene	138	108	78.26
IG_V_pseudogene	187	113	60.43
lincRNA	7,109	6,626	93.21
miRNA	3,049	1,177	38.6
miscRNA	2,033	1,165	57.3
Mt_rRNA	2	0	0
Mt_tRNA	22	22	100
Polymorphic pseudogene	45	43	95.56
Processed transcript	514	501	97.47
Protein-coding	20,327	19,970	98.24
Pseudogene	13,920	11,729	84.26
rRNA	526	0	0
Sense_intronic	741	684	92.31
Sense_overlapping	202	186	92.08
snoRNA	1,457	791	54.29
snRNA	1,916	925	48.28
TR_C_gene	5	5	100
TR_D_gene	3	0	0
TR_J_gene	74	28	37.84
TR_J_pseudogene	4	2	50
TR_V_gene	97	83	85.57
TR_V_pseudogene	27	21	77.78
TOTAL	57,773	49,129	85.04

Splicing issues

Given the confusion over the number of transcripts reliably detected, I was concerned about the claims made regarding splicing events. It is accepted that simple linear regression models, as used by the authors, is not a robust genome wide approach to identifying splice variants (See any review on splicing and genome wide methods) and while the authors focus on 5 candidates, it is very unclear how many of the ~1,100 “splice variants” they report are real.

[AU3] We understand that regression analysis is not a standard approach in this field, but choosing it was a thoughtful decision. We first screened and identified splicing mRNA variants and then we expressed them as percentages of the sums of all mRNA variants identified for a given gene. Percent expression was then used as the outcome variable for our regression analysis. Thus, regression is not used here to identify splicing variants but rather to test whether the percentage of each variant, in comparison with the total expression of variants (for a given gene) changes with aging. This approach is adopted in order to discriminate variability due to changes in transcription of a given gene from changes due to the expression of specific splicing variants. We acknowledge the limitations of using linear regression methods to investigate patterns in transcript isoform expression; however, building robust non-linear models to capture reliable trends requires a relatively large sample sizes and the chance of random results would be too high. We hope that clarifying the confusion between ENSGs and ENSTs solves the discrepancies raised by the reviewer.

We already know something is clearly wrong with tissue sequencing modelling, and this ‘something’ is well articulated in the literature e.g. data modelling and data quality issues - See Zhang C, Zhang B, Lin L-L, Zhao S. Evaluation and comparison of computational tools for RNA-seq isoform quantification. BMC Genomics. 2017;18(1):583, and citations within. In short, sequencing aspires to address splicing but so far, when applied to clinical tissue cohorts, it fails to live up to the hype. See Westoby et al 2020, for an update on the further issues with more advanced RNAseq methods.

The authors therefore need to reflect on the general failure of tissue RNAseq for splicing analysis e.g. Scott et al used RNAseq in a much larger muscle study than the present work (n=271) and yet the data quality is very poor. Despite claiming to be ‘transcriptome wide’ they report 1 splice candidate. Compare this with the performance of the splicing analysis presented by Sood et al 2016 (Nucleic Acids Res. 2016 Jun 20;44(11):e109. doi: 10.1093/nar/gkw263.) where thousands of splicing events are reliably detected across multiple human tissues (4,421 events). GTEx has a similar issue – reporting very little splicing across tissues (e.g. Mele 2016).

While there is no doubt sequencing (including 3’ & 5’ RACE) is critical for studying transcript variation in detail, the idea that its ideal for discovering the landscape of splicing in clinical cohorts is simply not yet correct. Furthermore, the tiling arrays have been able to study individual exons and splicing junctions for nearly a decade and so it is very misleading to claim that your RNAseq approach is doing something that was ‘not possible’ before.

[AU3] We recognize that the reviewer has a strong preference for using Affymetrix to perform splicing, and we were certainly able to find several studies in 2020 that used Affymetrix to investigate alternative splicing (PMIDs 32162363; 32078082). By contrast, the number of studies that used RNA-seq analysis to study splicing is far larger; just in 2020, there are dozens of such studies, including those with PMIDs 32554554; 32532357; 32503434; 32393171; 32314836; 32225167; 32175076; 32133419; 32123317; 32091665; 32043367; 31981701; 31960379; 31919425; 32117458; 31911676; 31911672; 31863578. We also found one study in 2020 in which both Affymetrix and RNA-seq were used (PMID 32211018). Thus, while

Affymetrix can certainly be used to study splicing, RNA-seq analysis is gaining popularity for splicing analyses.

We will not belabor the point about RNA-seq analysis being able to give insight into splicing that 'was not possible before', but tiling arrays, no matter how comprehensive, do not include all splicing junctions. Examples of such exceptions are circular RNAs, which arise from splicing and contain junctions that are not fully annotated, so these splicing events cannot be identified on arrays. Other examples include trans-splicing (splicing that ligates exons from different pre-mRNAs); these also are not fully annotated and are not represented in tiling arrays. We hope that this explanation alleviates the reviewer's concern.

Pathway analysis issues

Rev1]. Thanks - please note there is no significant Z-score and the p-value is weak. The threshold shown is not considered to be that useful for these pathway or ontology type analysis due to the inflated starting p-values that occur due to the database and sampling bias and the lack of independence of the variables (transcripts). For example, if you compared all detected gene symbols from your muscle study with the genome - you would see enormous p-values. So I would tread carefully with this particular result...

[AU2] Yes, we fully agree with the words of caution from the reviewer, and we added text to reflect this point in the revised manuscript.

If you fully agree you would remove anything that does not have a significant Z score? Given that you have so far not presented the correct values for the reliably 'detectable' transcriptome in your study you can not possibly have the correct transcriptomic background to feed into IPA and thus your statistics will not be reliable. You may even be missing events.

The correct list to submit to IPA as background is the gene symbols that represent the "transcript count is 24,543 on average". i.e. unless it is 1 transcript per gene in your analysis it will be a figure < 24,543.

[AU3] We have performed the Ingenuity pathway analysis as suggested by the reviewer using the gene symbols. Adipogenesis remains a significantly changed pathway. We are removing Suppl. Fig 9A and discussing this finding in the text.

Reviewers' Comments:

Reviewer #1:

Remarks to the Author:

As the authors are going off-topic in their rebuttal I have provided the key issues they have failed to address below and not in sequence to their rebuttal document. E.g. the number of arrays vs RNAseq carried out in the past year is irrelevant unless you are interested in commercial marketing. Your RNAseq study is clearly old (old methods) so was carried out many years ago. . There are still many 'claims' in their manuscript which are untrue or misrepresented, AND they are not required to publish this type of study so I do not see the need to include them.

I did not misunderstand the values you present – you continue to misspeak about the coverage you are achieving. Its really really simple. Your regression modeling must have used an input file that contain the subject ages and the genes COMMON to all subjects. Provide THAT number of Genes or transcripts (COMMON to every sample) and not numerical aggregates. Your new supplemental tables are therefore worse than misleading, they are disinformation. 24,453 genes detected in 1 or more samples is not meaningful way to refer to your data if you are modelling it vs a clinical variable. For example, did not reliably measure 6626 lincRNAs in all subjects – that value, which will be much lower, is the only parameter of interest for the regression analysis is the number COMMON across all samples that was then regressed vs age. You also use a low threshold for real detection, and that exaggerates your total numerical values. "We calculated that 24,453 ENSGs were expressed out of 57,773 ENSGs (the Ensembl v82 database)" – on AVERAGE.

But the final number that appeared in EVERY sample was? That figure is your 'reliable' transcriptome that you were then able to regress against age and falls far behind what has been achieved with tiling arrays (updated to the latest genome alignment – you have heard of custom CDFs?).

So you can't compare 22,887 (sood) with '24,453' (this study) as the former is a count detected in EVERY sample (and out-dated now) vs an AVERAGE NUMERICAL value that does not reflect the 'gene' detected in EVERY sample in your study.

"In this study, we found 68,349 unique protein-coding mRNA isoforms (ENSTs), and the corresponding proteins were enriched for pathways that are important in the context of aging. " That value is meaningless – what was the number of ENST detected in every sample and so entered into the regression analysis? If you don't have that number or have not used it, then you have not provided the correct background gene list for all of your ontology analysis – which would not be invalid.

Abstract

The abstract still contains multiple claims that are not true.

"Our study establishes a detailed framework of the global transcriptome and isoforms that govern muscle damage and homeostasis with age"

Their modelling does not account for variations in metabolic control (e.g. muscle insulin sensitivity) which vary due to diet or other genetic factors – and they have not studied muscle "damage".

Their study is a small replication study, carried out several years ago, that overlaps greatly with the biology of many previous publications. The authors would gain great credit and respect, if they were to present the study as such.

"Expression levels of 57,773 protein-coding and non-coding RNAs were studied as a function of aging by linear and negative binomial regression models"

This is also untrue. In line 526 of the methods you clearly state that "you detected 24,453.47 ± 4,610.29 RNAs at a coverage depth of ≥10 reads".

Even this number (24,453.47 ± 4,610.29) is not the number of RNAs that you detected in EVERY subject sample to then carry out the statistical modelling on. That will be a lower number and is the main number for comparing with the literature. By claiming you study 57,773 are counting

'events' that are not reliably presenting your data and that is not a transparent scientific approach.

Introduction

Line 53 to 57 should be removed as most 'claims' are either untrue or misrepresented, while they are not required to publish this type of replication study. E.g.

You have carried out "older" single-end RNA-seq and aligned to a 2015 genome build (a process known to have annotation challenges – see BMC Genomics, 2017 May 23;18(1):399. doi: 10.1186/s12864-017-3797-0. for an unbiased introduction to the issues that you are ignoring). Others have re-aligned exon/tiling array probe sequences to more updated genome builds, measured more RNA species (occurring in every sample), and thus have by any argument more accurate transcriptome data. This is not a useful argument for the importance of your project.

Secondly, e.g. the classification work of Sood et al – designed to distinguish very healthy young from very healthy old, matched for fitness as well as good general health. The same is true for many other muscle studies in the literature. Thus, the idea that the present small study represents a novel unique healthy population is not true. This issue is further compounded as you only consider age in your models and not factors, such as insulin resistance. The criteria for inclusion for literature studies is provided very clearly in the accompanying clinical articles. For example, in Timmons 2018 you will find plots for IR for every subject in each sub-set of data along with group values for aerobic fitness or blood pressure etc. You will also find statements in the recruitment descriptions of the clinical studies indicating that subjects had to not have any diagnosed clinical disease and not be on drug medication. Again, you appear to make your study sound 'special' when it is not.

Sample Preparation and Sequencing.

The authors continue – in an attempt to contemporise their data and results – to misrepresent the robustness of their 'RNA data'.

"Out of the 57,205 RNAs in the Ensembl hg19 v82 (September 2015) database". "To analyze RNA (ENSG) expression levels continuously with age, read counts for 57,205 RNAs were converted to log2-transformed counts per million"

This cannot be true. You did NOT measure 57,205 RNAs in EVERY subject. There must be a LOWER value common to all samples, that you then modelled. What is that number? It is not the numerical AVERAGE – that's a number that ignores the identity of the overlap.

Your regression modeling must have used an input file that contain the subject ages and the genes COMMON to all subjects. Surely you know what that number was? Or do you have a random sample size, where n varies depending on whether the gene was present .

Can the authors explain why they are using a reference transcriptome that is so old? Was their analysis carried out several years ago?

Illumina ChIP-Seq library prep kit – which one, please define. Impacts on the reliability of the technology.

Results and Nomenclature

Until the authors provide a simple .csv file that contains the normalised counts per ENST per subject it is impossible to get reasonable clarity on what data they actually modelled.

The authors have – in a valid attempt to clarify the details of the work – invented a strategy for naming molecules. There are only two valid terms they need to use: "genes" and "transcripts isoforms". The former can be measured at the DNA, RNA or protein level while the latter specifically refers to an RNA entity that is n = or > 1 per gene. There is no need to reinvent a language for a well-defined and well accepted nomenclature – and start calling RNA "ENSG" (that makes no sense as it's the nomenclature for a GENE).

"On average, $24,453 \pm 4,610$ RNAs were detected per sample in our study population, in which $15,291 \pm 1,759$ were protein-coding RNAs and $9,162 \pm 2,991$ were non-coding and other types of RNAs"

Average data is not used to carry out regression analysis – the important value is WHAT was the number of RNAs declared COMMON to all muscle samples, so that these COMMON values could then be regressed against Age. Give the HUGE variation (" $24,453 \pm 4,610$ RNAs") that value must be $\ll 24,453$ (and far less than the misleading value presented in the abstract). WHAT was that value and at what threshold for calling 'detected' was that achieved at? E.g. 1 RPKM ?? OR.....??

" $66,633 \pm 14,728$ isoforms, on average, were detected per sample in our study population, in which $27,908 \pm 4,527$ were protein-coding isoforms and $38,075 \pm 10,400$ were non-coding and other types of isoforms (Supplementary Fig. 1c). Overall, roughly 85% of all detected RNAs and 84% of all detected isoforms were identified in the Ensembl (hg19 v82) database, underscoring the reliability of our data (Supplementary Fig. 1b, c). "

As before, the standard deviations are enormous – and so claiming that you could detect "85% of the transcripts identified in Ensembl" is not only irrelevant for this type of study but its meaningless as you are regressing against clinical variables. Far deeper sequencing in muscle has been done.

Assuming 66,633 isoforms is where you are picking the 85% value from – does that mean that some samples had 140% of the Ensembl content while other samples had 40%?

I suggest they remove the splice variant section as its likely to be spurious given the massive variation in detectable transcriptome they have across subjects (which they struggle to acknowledge despite the data being clear and obvious in their standard deviations).

The number of significant events and the use of an FDR of 10% ensures this will be unreliable.

Citation errors and discussion

444. "In contrast to Phillips et al., 2013 , we identified several significantly enriched pathways linked to the top 506 differentially abundant RNAs. "

Phillips et al identified enriched numerous pathways

Comments on SKAP2. If it's a damage related event then it should co-vary with some other damage related markers – e.g. if its tissue histocytes then it would co-vary with some cell specific markers? That would be a novel analysis.

RESPONSE TO REVIEWER COMMENTS

Reviewer #1 (Remarks to the Author):

As the authors are going off-topic in their rebuttal I have provided the key issues they have failed to address below and not in sequence to their rebuttal document. E.g. the number of arrays vs RNAseq carried out in the past year is irrelevant unless you are interested in commercial marketing. Your RNAseq study is clearly old (old methods) so was carried out many years ago. There are still many 'claims' in their manuscript which are untrue or misrepresented, AND they are not required to publish this type of study so I do not see the need to include them.

[AU] We respectfully disagree, RNA-seq is not an 'old' method. The project started in 2015, and it took several years to recruit this very healthy aging cohort, especially participants over age 80. The careful selection of participants was necessary to look at the effect of aging only. As we mentioned in the previous revision, this approach is labor-intensive and time-consuming, but is critical for distinguishing as clearly as possible the effect of aging from disease. Sequencing proceeded without delay until the data were analyzed during 2019, and has been under review for almost one year now.

I did not misunderstand the values you present - you continue to misspeak about the coverage you are achieving. Its really really simple. Your regression modeling must have used an input file that contain the subject ages and the genes COMMON to all subjects. Provide THAT number of Genes or transcripts (COMMON to every sample) and not numerical aggregates.

Your new supplemental tables are therefore worse than misleading, they are disinformation. 24,453 genes detected in 1 or more samples is not meaningful way to refer to your data if you are modelling it vs a clinical variable. For example, did not reliably measure 6626 lincRNAs in all subjects - that value, which will be much lower, is the only parameter of interest for the regression analysis is the number COMMON across all samples that was then regressed vs age. You also use a low threshold for real detection, and that exaggerates your total numerical values.

"We calculated that 24,453 ENSGs were expressed out of 57,773 ENSGs (the Ensembl v82 database)" - on AVERAGE.

But the final number that appeared in EVERY sample was? That figure is your 'reliable' transcriptome that you were then able to regress against age and falls far behind what has been achieved with tiling arrays (updated to the latest genome alignment - you have heard of custom CDFs?).

So you can't compare 22,887 (sood) with '24,453' (this study) as the former is a count detected in EVERY sample (and out-dated now) vs an AVERAGE NUMERICAL value that does not reflect the 'gene' detected in EVERY sample in your study.

[AU] For linear regression model, we have used 57,205 genes (ENSGs) out of 57,773 genes in the database, including those genes from which protein-coding RNAs (mRNAs) are transcribed and those from which noncoding RNAs are transcribed only excluding the rRNA and genes which were not expressed in any sample. Given the heterogeneity of this cohort, if we had included only the common genes (>10 counts, as suggested by the reviewer) then we would have missed most genes encoding transcripts that change in abundance during aging, as some genes are not expressed at all in specific age groups. In this case, the number of shared genes that are expressed at >10 counts in all 53 samples is only 11,141. Including only transcripts that are detectable in all participants would have biased the analysis substantially. The levels of a transcript may be reduced below the threshold of detection at the extremes of the age distribution curve and therefore would be excluded; in fact, those genes may be

particularly important as they are affected by aging more than all the others. Therefore, we decided not to use this approach. We strongly believe that this is the most unbiased approach.

“In this study, we found 68,349 unique protein-coding mRNA isoforms (ENSTs), and the corresponding proteins were enriched for pathways that are important in the context of aging.”

That value is meaningless - what was the number of ENST detected in every sample and so entered into the regression analysis? If you don't have that number or have not used it, then you have not provided the correct background gene list for all of your ontology analysis - which would not be invalid.

[AU] Again, we respectfully disagree. The ENST values provide important information. We have used 196,354 ENSTs (transcripts) for regression modeling. Using only the common transcripts (>0.1 TPM) would have excluded transcripts showing altered expression with aging. For the reviewer's consideration, the number of common transcripts that are expressed (>0.1 TPM) in all 53 samples is 6,685.

Abstract

The abstract still contains multiple claims that are not true.

“Our study establishes a detailed framework of the global transcriptome and isoforms that govern muscle damage and homeostasis with age”

Their modelling does not account for variations in metabolic control (e.g. muscle insulin sensitivity) which vary due to diet or other genetic factors - and they have not studied muscle “damage”. Their study is a small replication study, carried out several years ago, that overlaps greatly with the biology of many previous publications. The authors would gain great credit and respect, if they were to present the study as such.

[AU] The reviewer may have missed that all participants were initially screened for insulin resistance both by measuring insulin and glucose at a fasting state in the morning and by performing a glucose tolerance test. By conceptualizing aging as a dynamic interaction between damage accumulation and repair mechanisms, we believe that studying aging, independent of disease, truly goes to the core of this issue, and has never been done before.

The comment about “respect” from this anonymous reviewer constitutes inappropriate and offensive judgement of our work. We do not intend in any way to compete with Timmons or his group, but rather contribute to understanding the effect of aging on muscle from multiple perspectives.

“Expression levels of 57,773 protein-coding and non-coding RNAs were studied as a function of aging by linear and negative binomial regression models”

This is also untrue. In line 526 of the methods you clearly state that “you detected $24,453.47 \pm 4,610.29$ RNAs at a coverage depth of ≥ 10 reads”.

Even this number ($24,453.47 \pm 4,610.29$) is not the number of RNAs that you detected in EVERY subject sample to then carry out the statistical modelling on. That will be a lower number and is the main number for comparing with the literature. By claiming you study 57,773 are counting “events” that are not reliably presenting your data and that is not a transparent scientific approach.

[AU] We regret that the reviewer continues to question the parameters we have used in our analysis of almost all genes (57,205 ENSGs), protein-coding and non-protein-coding only excluding rRNAs and genes which were not expressed in any sample. We have stated explicitly and transparently the parameters

we have used for analysis of these data. Please refer back to page 19 of the Methods section 'Regression Models and Visualization', which discusses our regression steps. Although the reviewer proposes that we analyze only the common genes (at >10 counts in all individuals), this parameter would have been too stringent for this cohort and we would have missed important genes differentially expressed with aging.

Introduction

Line 53 to 57 should be removed as most 'claims' are either untrue or misrepresented, while they are not required to publish this type of replication study. E.g. You have carried out "older" single-end RNA-seq and aligned to a 2015 genome build (a process known to have annotation challenges - see BMC Genomics, 2017 May 23;18(1):399. doi: 10.1186/s12864-017-3797-0. for an unbiased introduction to the issues that you are ignoring). Others have re-aligned exon/tiling array probe sequences to more updated genome builds, measured more RNA species (occurring in every sample), and thus have by any argument more accurate transcriptome data. This is not a useful argument for the importance of your project.

[AU] The GESTALT project started in March 2015. For comparing the results from many different aspects of the study, we decided to keep the genome annotation version 82 from Ensembl, which was released in September 2015 as per the Ensembl website. We understand the reviewer's preference, but like many other long-term projects, we fixed the genome version to what was available at the time we began our analysis.

Secondly, e.g. the classification work of Sood et al - designed to distinguish very healthy young from very healthy old, matched for fitness as well as good general health. The same is true for many other muscle studies in the literature. Thus, the idea that the present small study represents a novel unique healthy population is not true. This issue is further compounded as you only consider age in your models and not factors, such as insulin resistance. The criteria for inclusion for literature studies is provided very clearly in the accompanying clinical articles. For example, in Timmons 2018 you will find plots for IR for every subject in each sub-set of data along with group values for aerobic fitness or blood pressure etc. You will also find statements in the recruitment descriptions of the clinical studies indicating that subjects had to not have any diagnosed clinical disease and not be on drug medication. Again, you appear to make your study sound 'special' when it is not.

[AU] We respectfully disagree with these statements. First of all, insulin resistance was accounted for by our inclusion criteria. Insulin and fitness, as well as other criteria reported in previous papers, are certainly important, but are not as comprehensive as the rigorous criteria that were used in this study which included a complete medical evaluation for exclusion of any possible disease, no drug treatment, no mobility impairment, no clinical evidence of cardiovascular disease tested by both EKG and ultrasound, no serological or clinical evidence of serious infections, and a very extensive list of blood tests. We can repeat again the list of criteria, but it is already included in the manuscript.

Sample Preparation and Sequencing.

The authors continue - in an attempt to contemporise their data and results - to misrepresent the robustness of their 'RNA data'.

"Out of the 57,205 RNAs in the Ensembl hg19 v82 (September 2015) database". "To analyze RNA (ENSG) expression levels continuously with age, read counts for 57,205 RNAs were converted to log2-transformed counts per million"

This cannot be true. You did NOT measure 57,205 RNAs in EVERY subject. There must be a LOWER value common to all samples, that you then modelled. What is that number? It is not the numerical AVERAGE - that's a number that ignores the identity of the overlap.

Your regression modeling must have used an input file that contain the subject ages and the genes COMMON to all subjects. Surely you know what that number was? Or do you have a random sample size, where n varies depending on whether the gene was present.

[AU] As mentioned above, we have used almost all genes (57,205 ENSGs) out of 57,773 ENSGs in the database for the linear regression model excluding only rRNA genes and genes which were not expressed in any sample. Had we used only shared genes among all participants (>10 counts), the heterogeneity of this cohort would have caused an unfortunate loss of information on gene expression changes in aging muscle. In fact, since RNA-seq measures all transcripts, it is likely that 57,205 ENSGs is an underestimate. There could be many more which have not been annotated, as RNA-seq measures everything that is getting transcribed in a population of cells. The common gene number that is identified in all samples is 11,141 (>10 counts) but this has less biological value.

Can the authors explain why they are using a reference transcriptome that is so old? Was their analysis carried out several years ago?

[AU] As mentioned above, the GESTALT project started in March 2015, and for comparing the results from many different aspects of the study, we decided to keep the genome annotation version 82 from Ensembl, which was released in September 2015 as per Ensembl website. We understand the reviewer's preference but, as with other long-term projects, research teams can decide the genome version that is most sensible to follow, and we chose version 82, as it was the newest version available at the start of our study.

Illumina ChIP-Seq library prep kit - which one, please define. Impacts on the reliability of the technology.

[AU] We apologize for not adding this detail and have included it in the revised text. We used the TruSeq ChIP Library Preparation Kit, Set A (IP-202-1012) and Set-B (IP-202-1024), to prepare libraries from cDNA.

Results and Nomenclature

Until the authors provide a simple .csv file that contains the normalised counts per ENST per subject it is impossible to get reasonable clarity on what data they actually modelled.

[AU] Thank you for returning to this point. As we mentioned in the previous response letter and in the manuscript text, the files can be sent upon request. They are too large to upload in the *Nature Communications* website.

The authors have - in a valid attempt to clarify the details of the work - invented a strategy for naming molecules. There are only two valid terms they need to use: "genes" and "transcripts isoforms". The former can be measured at the DNA, RNA or protein level while the latter specifically refers to an RNA entity that is n = or > 1 per gene. There is no need to reinvent a language for a well-defined and well accepted nomenclature - and start calling RNA "ENSG" (that makes no sense as it's the nomenclature for a GENE).

[AU] With due respect, we have not invented a strategy for naming molecules. It is simply not correct to use the word 'genes' to refer to 'proteins', and although the field has been permissive with extending

the word 'gene' to 'RNA' in some cases, the relaxation of the word 'gene' has caused much confusion and ambiguity in gene expression literature. Hence, we stand by the strict use of 'gene' to refer to 'DNA' only. Exceptions to this rule, such as viruses with genetic information codified in RNA molecules, are outside of the purview of this study.

An 'RNA' or 'transcript' is, without ambiguity, the molecule that emerges through transcription of a gene, and a protein is, of course, the polypeptide synthesized from an mRNA. Transcript isoforms are differentially spliced RNAs transcribed from a given gene. As the reviewer mentioned earlier, referring to ENSTs, "Sood et al article reports >515,000 Exons detected while the available raw data and subsequent studies using the HTA 2.0 array with muscle tissue at GEO indicate they have transcript counts reliably detected in every sample of >100,000 ENSTs", we too used Ensembl nomenclature of ENSTs for human transcripts and ENSGs for human RNAs (genes).

"On average, 24,453 ± 4,610 RNAs were detected per sample in our study population, in which 15,291 ± 1,759 were protein-coding RNAs and 9,162 ± 2,991 were non-coding and other types of RNAs" Average data is not used to carry out regression analysis - the important value is WHAT was the number of RNAs declared COMMON to all muscle samples, so that these COMMON values could then be regressed against Age. Give the HUGE variation ("24,453 ± 4,610 RNAs") that value must be << 24,453 (and far less than the misleading value presented in the abstract). WHAT was that value and at what threshold for calling 'detected' was that achieved at? E.g. 1 RPMK ?? OR ...??

[AU] As mentioned previously, we have used almost all genes (57,205 ENSGs) in the database (57,773 ENSGs) for the linear regression model excluding only rRNA genes and those genes which were not expressed in any sample. Using only those genes present in all subjects (>10 counts) would have excluded important genes that may not be expressed at all in young persons or in older persons.

"66,633 ± 14,728 isoforms, on average, were detected per sample in our study population, in which 27,908 ± 4,527 were protein-coding isoforms and 38,075 ± 10,400 were non-coding and other types of isoforms (Supplementary Fig. 1c). Overall, roughly 85% of all detected RNAs and 84% of all detected isoforms were identified in the Ensembl (hg19 v82) database, underscoring the reliability of our data (Supplementary Fig. 1b, c)."

As before, the standard deviations are enormous - and so claiming that you could detect "85% of the transcripts identified in Ensembl" is not only irrelevant for this type of study but its meaningless as you are regressing against clinical variables. Far deeper sequencing in muscle has been done.

Assuming 66,633 isoforms is where you are picking the 85% value from - does that mean that some samples had 140% of the Ensembl content while other samples had 40%?

[AU] As mentioned before, we have used almost all genes (57,205 ENSGs; coding and non-coding) in the database (57,773 ENSGs) for the linear regression model excluding only rRNA genes and genes which were not expressed in any sample. With apologies for the repetition, had we followed the particular choice of this reviewer (i.e., including only those genes with RNAs expressed in all participants at >10 counts) we would have missed completely all those genes transcribed in low levels or totally not transcribed in certain age groups. As mentioned in the manuscript, when analyzing collectively all RNAs in all participants, we have identified roughly 85% of all RNAs in the Ensembl (hg19 v82) database.

I suggest they remove the splice variant section as its likely to be spurious given the massive variation in

detectable transcriptome they have across subjects (which they struggle to acknowledge despite the data being clear and obvious in their standard deviations).

The number of significant events and the use of an FDR of 10% ensures this will be unreliable.

[AU] We appreciate this suggestion, but the authors and the other reviewers consider the splice variant section as an integral part of the manuscript. In addition, an FDR of 10% is quite frequently used in genomic and transcriptomic studies and several articles published in *Nature Communications*: Cuomo et al., Nat Comm 11, 810, 2020) (<https://www.nature.com/articles/s41467-020-14457-z>) Schubert et al., Nat Comm 9, 20, 2018) (<https://www.nature.com/articles/s41467-017-02391-6>) Hausser et al., Nat Comm 10, 5423, 2019) (<https://www.nature.com/articles/s41467-019-13195-1>) Migliavacca et al., Nat Comm 10, 5808, 2019) (<https://www.nature.com/articles/s41467-019-13694-1>) Etc.

Citation errors and discussion

444. "In contrast to Phillips et al., 2013 , we identified several significantly enriched pathways linked to the top 506 differentially abundant RNAs. " Phillips et al identified enriched numerous pathways

[AU] We apologize for our miswording. "In contrast to" has been changed to "similar to" in the text.

Comments on SKAP2. If it's a damage related event then it should co-vary with some other damage related markers - e.g. if its tissue histocytes then it would co-vary with some cell specific markers? That would be a novel analysis.

[AU] We appreciate the reviewer's suggestion to expand on the analysis of SKAP2. Indeed, there were many interesting genes that deserved further analysis. However, due to limited sample availability and space constraints, we were not able to investigate covariance of genes in different pathways. We are expanding on this analysis in ongoing single-cell studies of skeletal muscle in different species.